# The COVID-19 pandemic and its impacts on diet quality and food prices in sub-Saharan Africa

Abbas Ismail[1], Isabel Madzorera[2,3], Edward A. Apraku[4], Amani Tinkasimile[5], Dielbeogo Dasmane[6], Pascal Zabre[7], Millogo Ourohire[7], Nega Assefa[8], Angela Chukwu[9], Firehiwot Workneh[10], Frank Mapendo[5], Bruno Lankoande[6], Elena Hemler[10], Dongqing Wang[10], Sulemana W. Abubakari[4], Kwaku P. Asante[4], Till Baernighausen[11], Japhet Killewo[12], Ayoade Oduola[13], Ali Sie[7], Abdramane Soura[7], Said Vuai[1], Emily Smith[14,15], Yemane Berhane[10], Wafaie W. Fawzi[3,16,17]

1 College of Natural and Mathematical Sciences, University of Dodoma, Dodoma, Tanzania, 2 Division of Community Health Sciences, School of Public Health, University of California, Berkeley, CA, United States of America, 3 Department of Global Health and Population, Harvard T.H. Chan School of Public Health, Harvard University, Boston, MA, United States of America, 4 Kintampo Health Research Center, Research and Development Division, Ghana Health Service, Kintampo, Bono East Region, Ghana, 5 Africa Academy for Public Health, Dar es Salaam, Tanzania, 6 Institut Supérieur des Sciences de la Population, University of Ouagadougou, Ouagadougou, Burkina Faso, 7 Nouna Health Research Center, Nouna, Burkina Faso, 8 College of Health and Medical Sciences, Haramaya University, Harar, Ethiopia, 9 Department of Statistics, University of Ibadan, Ibadan, Nigeria, 10 Addis Continental Institute of Public Health, Addis Ababa, Ethiopia, 11 Heidelberg Institute of Global Health, University of Heidelberg, Heidelberg, Germany, 12 Department of Epidemiology and Biostatistics, Muhimbili University of Health and Allied Sciences, Dar es Salaam, Tanzania, 13 University of Ibadan Research Foundation, University of Ibadan, Ibadan, Nigeria, 14 Department of Global Health, Milken Institute School of Public Health, George Washington University, Washington, DC, United States of America, 15 Department of Exercise and Nutrition Sciences, Milken Institute School of Public Health, George Washington University, Washington, DC, United States of America, 16 Department of Nutrition, Harvard T.H. Chan School of Public Health, Harvard University, Boston, MA, United States of America, 17 Department of Epidemiology, Harvard T.H. Chan School of Public Health, Harvard University, Boston, MA, United States of America

๏ These authors contributed equally to this work.
‡ YB and WWF also contributed equally to this work as last authors.
* imadzorera@berkeley.edu (IM); mina@hsph.harvard.edu (WWF)

**Data Availability Statement:** Individual participant data cannot be shared publicly. A data transfer agreement between Harvard T.H. Chan School of Public Health, Africa Academy for Public Health,

## Abstract

### Background

Sub-Saharan Africa faces prolonged COVID-19 related impacts on economic activity, livelihoods and nutrition, with recovery slowed down by lagging vaccination progress.

### Objective

This study investigated the economic impacts of COVID-19 on food prices, consumption and dietary quality in Burkina Faso, Ethiopia, Ghana, Nigeria, and Tanzania.

### Methods

We conducted a repeated cross-sectional study using a mobile platform to collect data from July-December, 2021 (round 2). We assessed participants' dietary intake of 20 food groups

and participating institutions (including Addis Continental Institute of Public Health, Nouna Health Research Center, Muhimbili University of Health and Allied Sciences, University of Dodoma, University of Ibadan, and Heidelberg Institute of Global Health) stipulates that data will be kept confidential and will not be shared beyond the research teams without prior permission. The de-identified dataset supporting this research may be made available following a request submitted to ghp@hsph.harvard.edu and be granted after obtaining permission from each participating institution.

**Funding:** This work was supported by institutional support from Harvard T.H. Chan School of Public Health, Boston, MA (WF); Harvard University Center for African Studies, Boston, MA (WF); Heidelberg Institute of Global Health, Germany (TB), and the George Washington University Milken Institute of Public Health, Washington, DC (ES). The funders have no role in study design, data collection and analysis, decision to publish, or preparation of the manuscript.

**Competing interests:** The authors have declared that no competing interests exist.

**Abbreviations:** ARISE, African Research, Implementation Science and Education Network; DDS, Dietary diversity score; DUCS, Dar es Salaam Urban Cohort Study; HDSS, Health and Demographic Surveillance System; MDD-W, Minimum Dietary Diversity for Women; PDQS, Prime Diet Quality Score; SSA, Sub-Saharan Africa; SSBs, sugar-sweetened beverages.

over the previous seven days and computed the primary outcome, the Prime Diet Quality Score (PDQS), and Dietary Diversity Score (DDS), with higher scores indicating better quality diets. We used generalized estimating equation (GEE) linear regression models to assess factors associated with diet quality during COVID-19.

## Results

Most of the respondents were male and the mean age was 42.4 (±12.5) years. Mean PDQS (±SD) was low at 19.4(±3.8), out of a maximum score of 40 in this study. Respondents (80%) reported higher than expected prices for all food groups. Secondary education or higher (estimate: 0.73, 95% CI: 0.32, 1.15), medium wealth status (estimate: 0.48, 95% CI: 0.14, 0.81), and older age were associated with higher PDQS. Farmers and casual laborers (estimate: -0.60, 95% CI: -1.11, -0.09), lower crop production (estimate: -0.87, 95% CI: -1.28, -0.46) and not engaged in farming (estimate: -1.38, 95% CI: -1.74, -1.02) were associated with lower PDQS.

## Conclusion

Higher food prices and lower diet quality persisted during the COVID-19 pandemic. Economic and social vulnerability and reliance on markets (and lower agriculture production) were negatively associated with diet quality. Although recovery was evident, consumption of healthy diets remained low. Systematic efforts to address the underlying causes of poor diet quality through transforming food system value chains, and mitigation measures, including social protection programs and national policies are critical.

## Introduction

The coronavirus disease 2019 (COVID-19) continues to affect the social, economic and health status of individuals and communities globally and has exposed significant inequalities by income, socio-demographic factors and geographic location [1]. Despite the persistent improvement of health globally, some settings continue to face greater threats to health and well-being during public health emergencies due to prevailing social, economic, political, and environmental conditions, and the COVID-19 pandemic is not an exception with greater impacts on the socially disadvantaged, including on the African continent [2, 3].

The absolute number of reported cases and mortality due to COVID-19 in Africa has been lower than in other regions, with 8.4 million cases and 170,300 deaths reported by March 2022 [4]. However, challenges with emerging variants are likely to continue in the region due to vaccine hesitancy and low vaccination coverage, in contrast to declining global cases and recovery of economies [5, 6]. Further, Sub-Saharan Africa (SSA) was already grappling with economic and health challenges before the pandemic, so the impacts of COVID-19 could be more long-lasting than in developed regions [7]. Poverty had been increasing globally before the COVID-19 pandemic and 768 million people were hungry in 2020 [8]. The African continent contributed more than one-third (282 million) of the hungry, and it has been projected that due to the COVID-19 pandemic, hunger will increase globally and even more in SSA [8]. Additionally, further increases in child stunting and wasting in SSA are anticipated [9].

The impact of COVID-19 on economic growth in SSA remains severe [8]. SSA economies are expected to continue to experience recession and slower economic growth, disruptions of

agriculture and production, and trade-related constraints if efforts to control COVID-19 remain limited [10]. Further, trade and fiscal deficits are likely to further adversely affect health and nutrition [11]. Countries and communities in SSA also continue to recover from COVID-19 related disruptions to livelihoods and access to key nutrition and food security services [12].

Prior to the COVID-19 pandemic, SSA food systems were already vulnerable to the influence of global markets, changing desirability, and poor food environments [13]. Studies early during the pandemic anticipated declines in food affordability, due to disruptions of food value chains, at the production, processing and distribution stages, and changes in consumer demand, with prices for major crops most affected [13, 14]. COVID-19 has also impacted vendors, markets and regulations with downstream effects on food availability, quality and prices [15]. Further, impacts on food security and dietary quality were anticipated due to market and manufacturing sector closures, restricted geographic access, and food price increases [13]. In our previous work in Burkina Faso, Ethiopia and Nigeria, we found evidence of increasing prices for key food groups, which may have contributed to lower dietary intake early during the COVID-19 pandemic [16]. We also found that higher pulse prices during the COVID-19 pandemic were associated with the consumption of less diverse diets [16]. It is important that as the COVID-19 pandemic persists we continue to assess its impact on the nutrition and health of Africans [12].

We investigate the continued impacts of COVID-19 on diet quality and food prices, using data collected from the African Research, Implementation Science, and Education (ARISE) Network cross-sectional COVID-19 studies in five SSA countries, Burkina Faso, Ethiopia, Ghana, Nigeria and Tanzania. This study contributes to understanding the indirect impacts of COVID-19 on diets in a region where data is limited. Data from several countries allows us to understand the diverse pathways through which COVID-19 affected nutrition. We also have repeated cross-sectional studies in 3 countries, and this allows us to track changes in food security and diet intake at different times during the COVID-19 pandemic.

## Material and methods

### Study setting

This study design was a repeated cross-sectional study with two rounds of data collection. The first round of the survey took place between July and November 2020. The study included six study sites from three countries, namely Nouna and Ouagadougou in Burkina Faso, Kersa and Addis Ababa in Ethiopia, and Ibadan and Lagos in Nigeria. In each country, one rural site and one urban site were selected. Rural sites included Nouna and Kersa, and a rural sub-area in Ibadan. As determined during the initial study design, a second-round survey was conducted to allow the assessment of the continuous effects of COVID-19 on many aspects of public health. The second survey took place between July and December 2021 and involved all sites involved in round 1 and an additional three sites from two countries, that is the rural sites of Kintampo in Ghana, the rural sub-area of Dodoma in Tanzania, and Dar es Salaam (urban) in Tanzania. The additional sites were included given additional access to resources and access to sites where ethical approval had taken longer to receive in round 1. The survey sites are shown in Fig 1.

The study countries and areas were selected based on existing collaborations and infrastructure available through the ARISE Network [17] and represent diverse settings across sub-Saharan Africa. More detailed information on the geographical and socio-demographic characteristics of the ARISE Network sites has been provided elsewhere [17, 18].

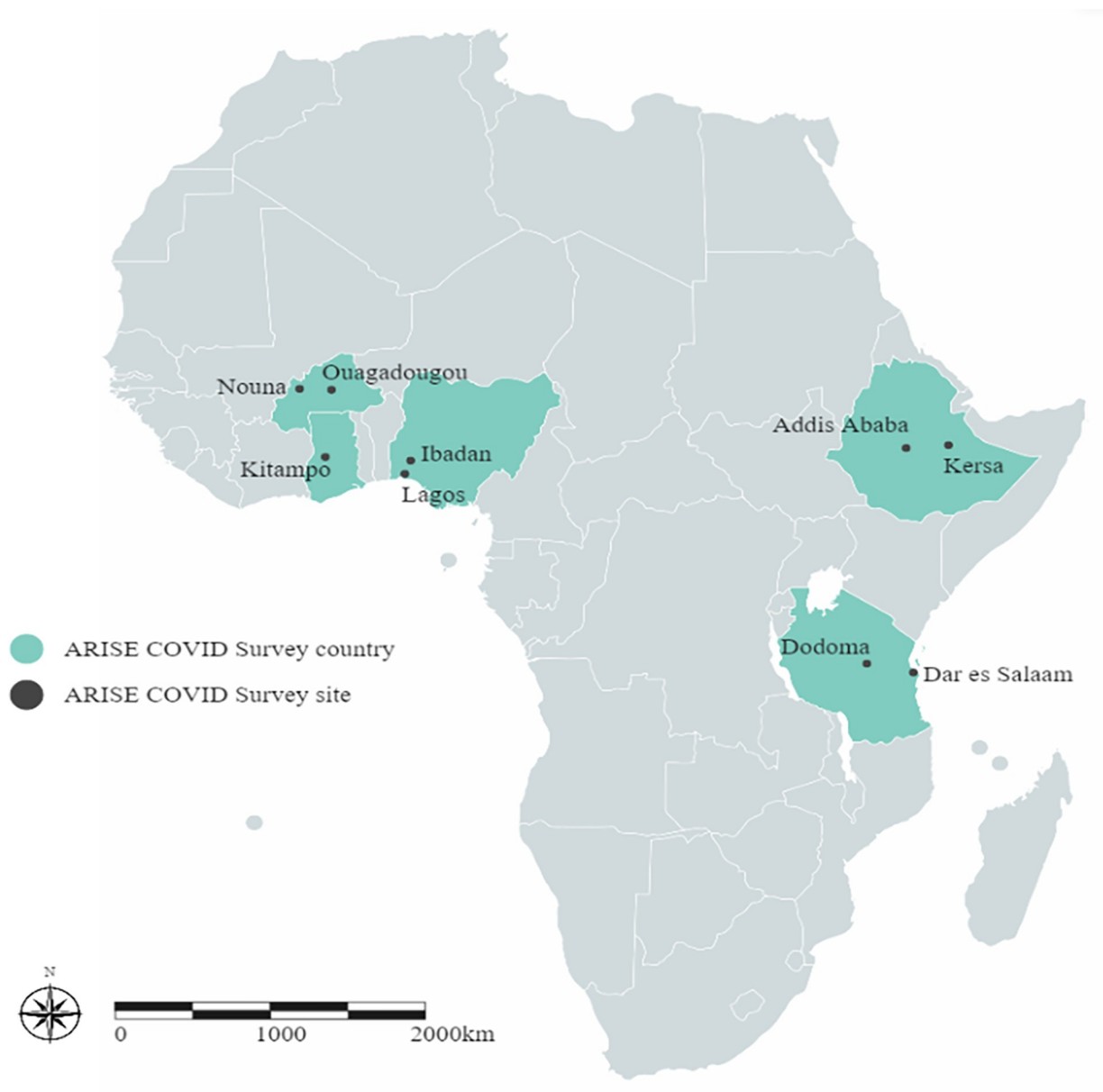

**Fig 1. Map of ARISE sites for Round 2 of the COVID-19 studies.**

## Study design

The study used a mobile phone platform and computer-assisted telephone interviewing (CATI) to collect data. Research assistants obtained verbal informed consent before starting the interview. They conducted interviews from study call centers.

Each country surveyed a minimum of 300 healthcare workers, 600 adults and 600 adolescents (except Ghana, where we surveyed 300 adults and 300 adolescents) in round 2. This analysis involved data from the adult surveys. We randomly selected a larger number of phone numbers of households from the sampling frame, to allow for non-response, refusal or dropout. For the household surveys, 2,500 households were selected from each urban and rural site in Round 1, assuming that 60% of the households would respond to the survey. The

assumption allowed us to reach the minimum number of 300 adults per site. From each household, an adult aged 20 years or above was selected for the interview.

The second round of data collection targeted the same adults previously surveyed in Round 1 (Burkina Faso, Ethiopia and Nigeria) and additional new respondents to account for non-response and drop-outs. For new countries, the sample was obtained from existing sampling frames. The sampling frames were obtained from existing surveillance systems, including the Health and Demographic Surveillance Systems (HDSS) in Burkina Faso, Ghana, Nigeria and rural Ethiopia (Kersa), and from a household survey in Addis Ababa (Ethiopia) that was established in the first round of data collection. In Tanzania, the sampling frames were obtained from the Dar es Salaam Urban Cohort Study (DUCS) and HDSS and the Dodoma HDSS. Table 1 shows the number of participants in surveys 1 and 2. We ensured consistency in study design and questions across all sites to ensure that differences between the two rounds would not cause issues in comparisons (across countries and rounds if applicable) in the analysis.

The detailed design and field methods of the round 1 ARISE Network COVID-19 rapid monitoring survey have been published elsewhere [18]. The design and methods of the round 2 survey are detailed on the Harvard University Center for African Studies website [19].

Standardized tools used for the first survey round were updated for the second survey round. The local teams from each site reviewed the tools and updated the questions accordingly. Questionnaires were translated into the local languages. The sites recruited male and female research assistants who then received extensive training on data collection procedures, including the use of telephones and tablets to obtain verbal informed consent and input participant data into an electronic data collection system (Open Data Kit). Research assistants collected information on socio-demographic characteristics, including age, sex, head of household, household size, education, and occupation of respondents. Questions regarding food pricing, food security, and the dietary intake of respondents were also asked.

## Outcome variables

The primary outcome of interest for this study was the Prime Diet Quality Score (PDQS). We also computed the diet diversity score (DDS) as a secondary outcome. The computation of the dietary indices, the PDQS, and DDS considered both Rounds of data collection. Briefly, respondents were asked about their frequency of consumption of 20 food groups commonly consumed in study areas over the preceding seven days. Respondents were asked to recall the number of days they consumed food from a list of 20 food groups over the past 7 days period before the COVID-19 emergency and during the COVID-19 pandemic (in the round 1 survey) [20] and in round 2 we asked respondents to recall their consumption of the same 20 food groups over the previous 7 days.

**Prime diet quality score (PDQS).** We computed the PDQS, a measure of diet quality based on reported dietary intake. Previous studies have found associations between PDQS and birth outcomes, pregnancy-related morbidities, diabetes and cardiovascular disease [21–24]. In a study using an alternative classification of the PDQS in the United States, the PDQS was negatively associated with food insecurity [25]. We classif d from the 20 food groups foods into 14 healthy food groups (dark green leafy vegetables, other vitamin A-rich vegetables including carrots, cruciferous vegetables, other vegetables, whole citrus fruits, other fruits, fish, poultry, legumes, nuts, low-fat dairy, whole grains, eggs and liquid vegetable oils) and 6 unhealthy food groups (red meat, processed meats, refined grains and baked goods, sugar-sweetened beverages (SSBs), desserts and ice cream and fried foods obtained away from home and potatoes) based on PDQS criteria determined by previous studies [21, 23]. Points were assigned for consumption of healthy food groups as: 0–1 serving/week (0 points), 2–3

**Table 1. Demographic characteristics of the study individuals and households in Burkina Faso, Ethiopia, Nigeria, Tanzania and Ghana (N = 2,829).**

| | Overall | Burkina Faso | | Ethiopia | | Nigeria | | Tanzania | | Ghana |
|---|---|---|---|---|---|---|---|---|---|---|
| | | Nouna | Ouagadougou | Kersa | Addis Ababa | Ibadan | Lagos | Dar es Salaam | Dodoma | Kintampo |
| Location | | Rural | Urban | Rural | Urban | Rural | Urban | Urban | Rural | Rural |
| N | 2829 | 324 | 300 | 298 | 289 | 373 | 290 | 307 | 347 | 301 |
| *Sociodemographic characteristics* | | | | | | | | | | |
| Female sex | 1232 (43.6) | 50(15.4) | 103(34.3) | 44(14.8) | 202(69.9) | 164(44.0) | 131 (45.2) | 146(47.6) | 232(66.9) | 160(53.2) |
| Age of respondent (Mean ±SD) years | 42.4±12.5 | 47.6 ±12.8 | 47.4±10.1 | 36.6±9.4 | 38.0±12.2 | 40.4 ±13.1 | 39.5 ±11.8 | 48.6±12.5 | 43.0 ±10.1 | 40.0±13.6 |
| 20–29 | 428 (15.1) | 15(4.6) | 8(2.7) | 66(22.1) | 72(24.9) | 88(23.6) | 55(19.0) | 16(5.2) | 32(9.2) | 76(25.3) |
| 30–39 | 764 (27.0) | 66(20.4) | 52(17.4) | 116 (38.9) | 112(38.8) | 91(24.4) | 99(34.1) | 53(17.3) | 96(27.7) | 79(26.3) |
| ≥ 40 | 1636 (57.9) | 243 (75.0) | 239(79.9) | 116 (38.9) | 105(36.3) | 194(52.0) | 136 (46.9) | 238(77.5) | 219(63.1) | 146(48.5) |
| Education | | | | | | | | | | |
| None or incomplete primary | 1060 (37.9) | 254 (79.3) | 220(74.1) | 207 (69.5) | 83(28.7) | 20(5.5) | 5(1.8) | 15(4.9) | 112(32.3) | 144(47.8) |
| Primary school or incomplete secondary | 826 (29.6) | 53(16.6) | 60(20.2) | 61(20.5) | 77(26.6) | 51(14.1) | 18(6.5) | 211(69.2) | 229(66.0) | 66(21.9) |
| Secondary school or higher | 909 (32.5) | 13(4.1) | 17(5.7) | 30(10.1) | 129(44.6) | 290 (80.3) | 254 (91.7) | 79(25.9) | 6(1.7) | 91(30.2) |
| *Household* | | | | | | | | | | |
| Head of Household | 1868 (66.0) | 244 (75.3) | 237(79.0) | 265 (88.9) | 211(73.0) | 187(50.1) | 155 (53.5) | 224(73.0) | 184(53.0) | 161(53.5) |
| Household size (Mean ±SD) | 6.4±3.9 | 10.4±5.9 | 7.9±3.3 | 6.7±2.6 | 4.7±2.4 | 4.7±2.8 | 4.4±2.7 | 5.5±2.5 | 6.3±2.1 | 7.3±3.9 |
| Occupation | | | | | | | | | | |
| Unemployed | 318(12.2) | 18(6.2) | 56(22.3) | 22(7.4) | 118(40.8) | 15(5.2) | 4(1.6) | 50(16.3) | 2(0.6) | 33(11.1) |
| Farmer or casual labor | 1027 (39.4) | 239 (82.7) | 32(12.8) | 250 (84.5) | 1(0.4) | 11(3.8) | 5(2.0) | 24(7.8) | 342(98.6) | 123(41.3) |
| Employed | 418(16.0) | 13(4.5) | 35(13.9) | 14(4.7) | 59(20.4) | 100(34.7) | 103 (41.9) | 49(16.0) | 4(0.6) | 43(14.4) |
| Student, self-employed or other | 847 (32.5) | 19(6.6) | 128(51.0) | 10(3.4) | 111(38.4) | 162(56.3) | 134 (54.5) | 183(59.8) | 1(0.3) | 99(33.2) |
| Religion | | | | | | | | | | |
| Catholic | 355(12.6) | 94(29.0) | 83(27.7) | 0(0.0) | 2(0.7) | 22(6.0) | 12(4.2) | 88(28.7) | 20(5.8) | 34(11.3) |
| Muslim | 1185 (42.0) | 203 (62.7) | 201(67.0) | 291 (97.7) | 36(12.5) | 110(29.9) | 60(20.8) | 153(49.8) | 0(0.0) | 131(43.5) |
| Orthodox Christian | 653(23.2) | 1(0.3) | 0(0) | 6(2.0) | 227(79.0) | 175(47.6) | 205 (71.2) | 6(2.0) | 12(3.5) | 21(7.0) |
| Protestant | 5858 (20.0) | 23(7.1) | 15(4.9) | 1(0.3) | 19(7.3) | 54(14.7) | 6(2.1) | 59(19.2) | 311(89.9) | 97(32.2) |
| None or other | 40(1.4) | 3(0.9) | 1(0.3) | 0(0) | 2(0.7) | 7(1.9) | 5(1.7) | 1(0.3) | 3(0.9) | 18(6.0) |
| PDQS | 19 | 19 | 18 | 18 | 18 | 21 | 22 | 20 | 18 | 19 |
| (Median, IQR) | (17, 22) | (17, 21) | (16, 21) | (15, 20) | (15, 20) | (19, 24) | (20, 24) | (17, 22) | (16, 20) | (16, 22) |

Data are shown as mean ±SD or N (percent), IQR: Interquartile range

servings/week (1 point), and ≥4 servings/week (2 points). Scoring for unhealthy food groups was assigned as: 0–1 serving/week (2 points), 2–3 servings/week (1 point) and ≥4 servings/week (0 points). Points for each food group were then summed to give an overall score (maximum score of 40).

**Dietary diversity score (DDS).** We also computed the Dietary diversity score (DDS) based on the classification suggested for the Food and Agriculture Organization [FAO]'s Minimum Dietary Diversity for Women (MDD-W) index. The MDD-W has been validated and is considered a good measure of micronutrient adequacy among women [26, 27]. We grouped the food consumed by study participants into 10 food groups as follows: 1) grains, white roots and tubers and plantains, 2) legumes (beans, peas and lentils), 3) nuts and seeds, 4) dairy, 5) meats, poultry and fish, 6) eggs, 7) vitamin A rich dark green vegetable, 8) other vitamin A rich fruits and vegetables, 9) other vegetables, and 10) other fruits. We divided the reported weekly consumption of the food groups by seven to obtain the daily frequency of consumption. If the food was eaten at least once daily during the previous week, it was considered to contribute to DDS, with a higher number indicating higher dietary diversity.

## Exposure variables

We considered as exposures of interest changes in food pricing by comparing the time before and late during the COVID-19 pandemic for staples (maize, rice, cassava and teff), pulses (beans, lentils, peas, chickpeas), fruits (e.g. bananas, oranges, any locally available fruits), vegetables (e.g. spinach, cabbage, tomatoes, onions, any locally available vegetables) and animal source foods (e.g. beef, chicken, dairy, eggs, fish). We created a binary indicator indicating an increase compared to a decrease or no change in food prices (yes or no). The changes in food pricing were determined before and after March 2020 (first round), when the first case of COVID-19 was reported and also in the second round. Changes in food prices were self-reported by the participants.

We also considered food security status (e.g., went without eating for a whole day), and the impact of COVID-19 on income, employment and crop production. We considered respondent age (20–29, 30–39, $\geq$ 40 years), sex (female or male), education status (none or incomplete primary, primary school or incomplete secondary, secondary school or higher), respondent is head of household (yes or no), occupation (unemployed, farmer or casual laborer, employed, student, self-employed or other), religion (none, Catholic, Muslim, Orthodox Christian, Protestant or other). Other exposures included household characteristics such as household size and a wealth index score (tertiles). We computed the wealth index based on factor analysis on ownership of common household assets and other wealth-related indicators such as donkey cart, radio, television, bicycle, motorcycle ownership, access to grid electricity, improved water, fuel and roof within each country. We classified respondents in each country into wealth tertiles based on results from the country specific factor analysis findings.

## Statistical analysis

Descriptive statistics were used to describe numerical, tabular and graphical presentations of the data. We used means and standard deviations for continuous data and frequencies and percentages for categorical data to summarize social demographic characteristics, nutrition and food security across sites and rounds. We used Generalized Estimating Equation (GEE) linear regression models [28], to assess factors associated with the PDQS at the round 2 endpoint.

For those respondents with both round 1 and round 2 data, we evaluated the factors affecting diet quality in round 2 of the study. We conducted sensitivity analysis and assessed among respondents with both rounds 1 and 2, if adjusting for diet quality before the COVID-19 pandemic would change observed associations. We used GEE linear regression models to assess factors associated with the PDQS at the round 2 endpoint accounting for diet quality at round 1.

Predictors of dietary quality were determined based on univariate selection. Variables associated with the outcomes at p<0.20 were included in the model. The final models include all selected covariates. Significance was determined at p<0.05. We used SAS version 9.4 for all analyses.

### Ethical approval and consent

The study obtained ethical approval from the Institutional Review Board at Harvard T.H. Chan School of Public Health, the Kintampo Health Research Centre Institutional Ethics Committee (Ghana), Nouna Health Research Center Ethical Committee and National Ethics Committee (Burkina Faso), the Institutional Ethical Review Board of Addis Continental Institute of Public Health (Ethiopia), University of Ibadan Research Ethics Committee and National Health Research Ethics Committee (Nigeria), and the Muhimbili University of Health and Allied Sciences and National Institute for Medical Research (Tanzania).

## Results

We analyzed data from 2,829 adults from five countries. Overall, the study included 44% female respondents. Female participation was low in Nouna (15.4%), Ouagadougou (34.3%) and Kersa (14.8%). The mean (±SD) age of the respondents was 42.4 (±12.5) years. Most of the study respondents had no education or incomplete primary school education (37.9%), and 32.5% had secondary school education or higher. The mean (±SD) household size was 6 (±4) people. Most respondents were farmers or casual laborers (39.4%) and 32.5% were students or self-employed. A more detailed description of the demographic characteristics of the study individuals can be found in **Table 1**.

### Changes in income, employment, crop production and food security status

**Table 2** shows the impact of COVID-19 on various factors that affect diet quality in the round 2 survey. Most respondents (49%) reported no change in income, and 37% reported loss or reduced income from farming, entrepreneurial activities, or business activities during the COVID-19 pandemic. Reported income losses from agriculture, entrepreneurial activities, or formal or informal business were highest in Kintampo (58%), Dar es Salaam (50%), and Ibadan (42%). Lost or reduced salaries were reported most in Ouagadougou (37%) and Lagos (26%). The majority of respondents, however, reported that COVID-19 had not affected their employment status (only 7% reported lost employment). Loss of employment was relatively higher in Addis Ababa (16%), Ouagadougou (14%) and Ibadan and Nouna (at least 10%). Approximately 19% of all respondents across all sites reported lower agriculture production during the COVID-19 pandemic, with the highest declines reported in the rural sites of Kersa (64%), Nouna (37%) and Kintampo (27%). Food insecurity also affected respondents, with 45% reporting that they worried about food, and 30% said they had skipped a meal. Almost 10% had gone for an entire day without eating. These three questions are components of a metric for the assessment of food insecurity, the Household Food Insecurity Access Scale (HFIAS). The number of people reporting skipping a meal was highest in Ibadan (57%), Lagos (52%) and Kintampo (46%). Going for an entire day without eating was reported in Lagos (20%) and Ouagadougou (17%). Finally, social protection was limited, with less than 1% of the study sample reporting access to food aid, and 2% reporting access to cash transfers (results not shown in tables or figures).

Respondents in rural regions of all sites were less likely to report changes in income or loss of salary compared to those residing in urban areas. Rural respondents in Ethiopia and Tanzania were also less likely to report reduced income, however, in other sites there were no rural-

**Table 2. Impact of COVID-19 on income, employment, crop production, and food security across five countries in the Round 2 study.**

| | Overall | Burkina Nouna | Burkina Ouaga | Ethiopia Addis | Ethiopia Kersa | Nigeria Ibadan | Nigeria Lagos | Tanzania Dar es Salaam | Tanzania Dodoma | Ghana Kintampo |
|---|---|---|---|---|---|---|---|---|---|---|
| Income is unchanged | 49.4 | 61.6 | 29.7 | 48.1 | 82.1 | 43.3 | 37.7 | 40.9 | 78.1 | 34.8 |
| Lost/reduced salary | 12.1 | 4.3 | 36.8 | 9.1 | 0.7 | 11.9 | 26.1 | 7.9 | 0.6 | 6.4 |
| Lost/reduced income | 36.8 | 34.1 | 33.5 | 36.6 | 16.9 | 41.9 | 33.8 | 49.6 | 21.3 | 57.9 |
| Increased salary | 1.7 | 0.0 | 0.0 | 6.3 | 0.3 | 2.8 | 2.5 | 1.7 | 0.0 | 1.0 |
| **COVID-19 impact on employment** | | | | | | | | | | |
| None, unemployed | 31.1 | 46.8 | 30.0 | 43.9 | 10.4 | 16.5 | 13.6 | 41.0 | 55.9 | 20.7 |
| None, no change in employment status | 52.6 | 42.4 | 44.6 | 35.0 | 83.6 | 62.3 | 71.0 | 21.5 | 39.7 | 68.9 |
| Lost employment | 7.1 | 10.2 | 13.5 | 15.9 | 0.7 | 10.6 | 3.9 | 3.9 | 1.2 | 4.4 |
| Changed occupation | 3.1 | 0.3 | 2.3 | 4.8 | 1.7 | 3.3 | 5.2 | 4.3 | 1.5 | 5.0 |
| Other (specify) | 6.2 | 0.3 | 9.7 | 0.4 | 3.7 | 7.3 | 6.3 | 29.3 | 1.7 | 1.0 |
| **Crop production affected by COVID-19** | | | | | | | | | | |
| No | 36.6 | 44.3 | 23.7 | 4.2 | 32.7 | 6.5 | 31.4 | 67.4 | 91.9 | 22.2 |
| Yes, production has decreased | 18.6 | 36.6 | 9.4 | 1.0 | 63.6 | 11.1 | 7.7 | 4.6 | 8.1 | 26.9 |
| Yes, production has increased | 1.8 | 1.0 | 3.2 | 1.4 | 0.3 | 3.5 | 1.4 | 0.7 | 0.0 | 4.7 |
| No, I don't grow any crops | 43.0 | 18.2 | 63.7 | 93.4 | 3.4 | 78.9 | 59.6 | 27.4 | 0.0 | 46.3 |
| **Food insecurity** | | | | | | | | | | |
| Worried about food | 45.4 | 22.5 | 47.0 | 50.2 | 82.6 | 59.3 | 52.3 | 21.5 | 19.7 | 57.7 |
| Skipped a meal | 30.3 | 2.5 | 33.6 | 28.4 | 21.5 | 56.7 | 55.6 | 14.7 | 14.1 | 45.5 |
| Went for entire day without eating | 9.8 | 2.2 | 17.0 | 11.8 | 4.7 | 10.5 | 20.3 | 2.9 | 8.7 | 12.0 |

urban differences or income reductions affected a greater proportion of rural respondents. Across all sites except Nigeria, rural respondents were more likely to report no change in employment status. In rural sites more people reported worrying about food (except in Dodoma and Nouna), however, going for an entire day without eating was more frequently reported in urban sites (except in Tanzania).

Among respondents that were in rounds 1 and 2, reports of disruptions to agricultural production were more prevalent in round 1 as at least 44% indicated that crop production had decreased. Additionally, more respondents had worried about running out of food (57%) and fewer had skipped a meal (21%) in round 1 (results not shown in tables or figures).

## Changes in prices for key food groups

In round 2 of the study, respondents reported that for all food groups, prices reported later during the COVID-19 pandemic were higher than typical prices in the previous year (Fig 2). Higher than expected prices were noted for all food groups across all sites for at least 80% of respondents, except in Dar es Salaam and Dodoma.

Among respondents that were in rounds 1 and 2, at least 82% had reported increases in staple prices in round 1 compared to the time before the COVID-19 pandemic. For pulses, 77% reported increased prices, for fruits 66% reported higher prices, and 74% and 73% for vegetables and animal foods respectively. For all food groups except vegetables, higher prices were most frequently reported in round 1 (early during the COVID-19 pandemic) compared to

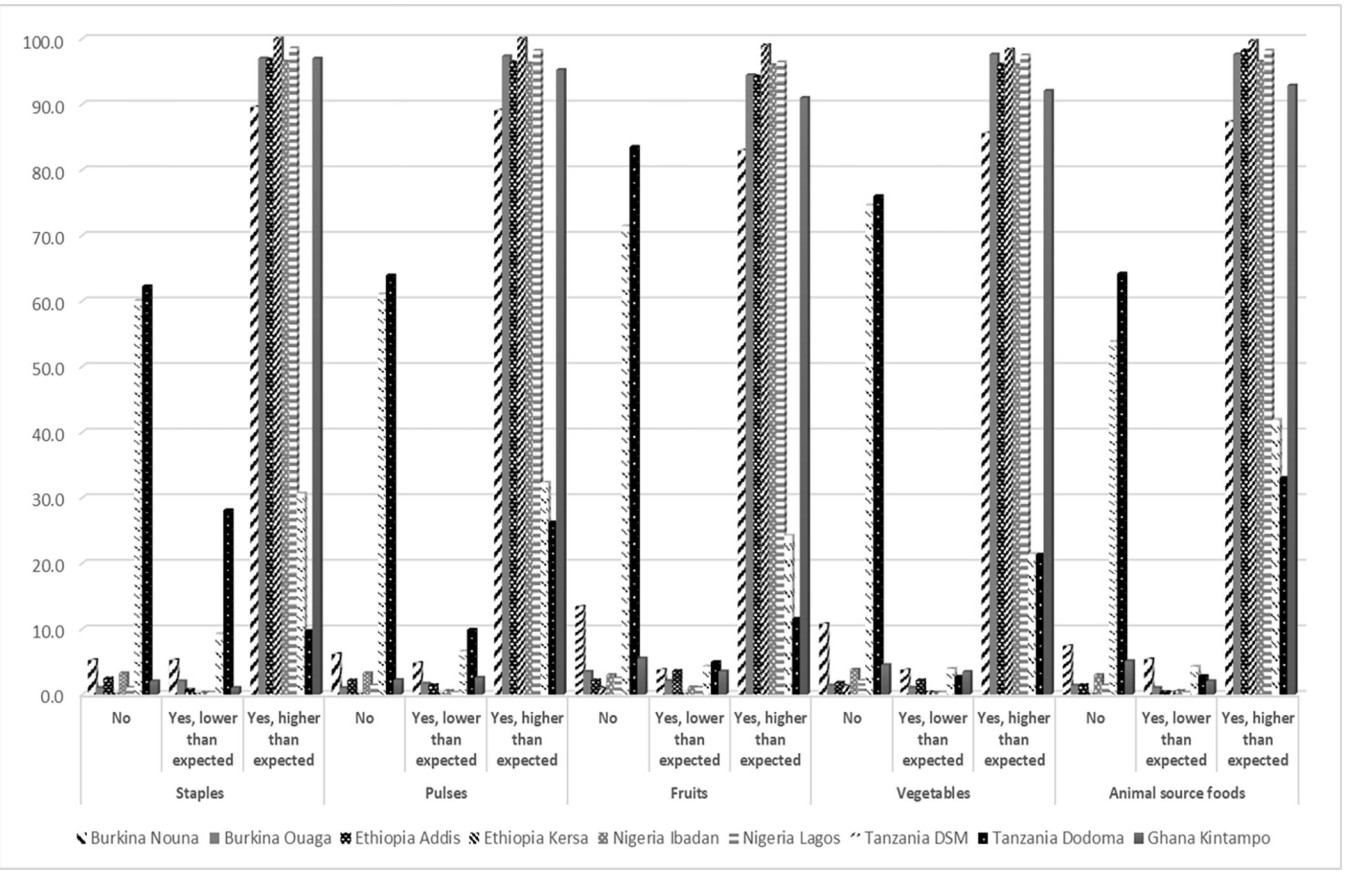

**Fig 2. The proportion of people reporting changes in prices of key food groups during the COVID-19 pandemic, as compared to this time of the year in previous years.**

round 2 (results not shown in tables or figures). The food groups where there were the largest differences in prevalence of higher prices were fruits and animal-source foods.

## Effects on food consumption and diet quality

**Consumption of healthy foods.** Table 3 shows the frequency of consumption of healthy PDQS food groups across study sites in the five countries (Round 2). Consumption of dark green vegetables was high in Dar es Salaam, Kintampo and Dodoma, where 62%, 44%, and 38% of the respondents reported consuming these food groups four or more times a week, respectively. However, in Ethiopia, at least 69% reported consumption of dark green vegetables on one day or not at all in the previous week. Consumption of other vitamin A-rich vegetables was equally low in Burkina Faso, Ethiopia, Kintampo and Dodoma. Respondents reported low consumption of citrus and other fruits across most sites, except in Dar es Salaam, Ibadan, Lagos and Addis Ababa, where at least 22% of respondents reported citrus fruit consumption four or more times a week. In Addis Ababa, Kersa and Dodoma, most respondents reported consuming fish only once or not at all in the previous week. Poultry consumption was also low across the majority of sites, with more than 70% of respondents reporting consumption less than twice in the prior week, although intake was relatively higher in Lagos and Kintampo. In Addis Ababa (46%), Dar es Salaam (40%) and Dodoma (37%), respondents reported legume consumption most frequently (≥4 times a week). Whole-grain intake was higher in rural sites

**Table 3. Frequency of consumption of healthy PDQS food groups in Burkina Faso, Ethiopia, Nigeria, Tanzania and Ghana in the Round 2 study.**

| | Frequency (%) | Burkina Nouna | Burkina Ouaga | Ethiopia Addis | Ethiopia Kersa | Nigeria Ibadan | Nigeria Lagos | Tanzania Dar es Salaam | Tanzania Dodoma | Ghana Kintampo |
|---|---|---|---|---|---|---|---|---|---|---|
| Dark green vegetables | < = 1 per week | 31.4 | 48.7 | 86.5 | 69.1 | 18.2 | 22.8 | 3.9 | 16.1 | 28.6 |
| | 2–3 times per week | 52.2 | 43.7 | 11.1 | 24.2 | 55.2 | 50.0 | 33.9 | 45.8 | 27.6 |
| | 4 or more times per week | 16.4 | 7.7 | 2.4 | 6.7 | 26.5 | 27.2 | 62.2 | 38.0 | 43.9 |
| Other Vit A vegetables | < = 1 per week | 60.2 | 68.7 | 60.2 | 77.5 | 20.4 | 36.9 | 26.1 | 49.3 | 67.8 |
| | 2–3 times per week | 37.0 | 25.7 | 29.8 | 20.5 | 58.7 | 42.8 | 45.9 | 31.7 | 20.9 |
| | 4 or more times per week | 2.8 | 5.7 | 10.0 | 2.0 | 20.9 | 20.3 | 28.0 | 19.0 | 11.3 |
| Cruciferous vegetables | < = 1 per week | 45.4 | 41.3 | 63.7 | 40.9 | 48.8 | 43.8 | 48.2 | 62.0 | 71.8 |
| | 2–3 times per week | 51.5 | 48.7 | 31.1 | 42.3 | 35.7 | 36.2 | 42.7 | 28.2 | 16.3 |
| | 4 or more times per week | 3.1 | 10.0 | 5.2 | 16.8 | 15.6 | 20.0 | 9.1 | 9.8 | 12.0 |
| Other vegetable | < = 1 per week | 9.9 | 9.7 | 2.1 | 6.4 | 4.8 | 7.9 | 15.3 | 18.7 | 2.0 |
| | 2–3 times per week | 49.1 | 39.3 | 2.1 | 14.4 | 30.6 | 30.3 | 30.6 | 38.3 | 6.3 |
| | 4 or more times per week | 41.1 | 51.0 | 95.9 | 79.2 | 64.6 | 61.7 | 54.1 | 42.9 | 91.7 |
| Citrus | < = 1 per week | 65.1 | 66.3 | 54.7 | 89.6 | 28.4 | 28.6 | 42.4 | 85.0 | 73.8 |
| | 2–3 times per week | 32.4 | 26.0 | 22.8 | 9.4 | 46.1 | 44.1 | 35.5 | 11.2 | 14.6 |
| | 4 or more times per week | 2.5 | 7.7 | 22.5 | 1.0 | 25.5 | 27.3 | 22.2 | 3.8 | 11.6 |
| Other fruit | < = 1 per week | 57.5 | 69.7 | 65.4 | 89.9 | 30.6 | 34.1 | 22.5 | 74.6 | 70.8 |
| | 2–3 times per week | 40.1 | 25.3 | 24.9 | 10.1 | 52.8 | 43.1 | 41.7 | 22.5 | 19.3 |
| | 4 or more times per week | 2.5 | 5.0 | 9.7 | 0.0 | 16.6 | 22.8 | 35.8 | 2.9 | 10.0 |
| Fish | < = 1 per week | 5.3 | 20.3 | 99.0 | 99.0 | 4.8 | 12.1 | 47.6 | 76.7 | 23.3 |
| | 2–3 times per week | 56.2 | 44.7 | 1.0 | 0.7 | 31.1 | 25.2 | 39.4 | 17.8 | 18.3 |
| | 4 or more times per week | 38.6 | 35.0 | 0.0 | 0.3 | 64.1 | 62.8 | 13.0 | 5.5 | 58.5 |
| Poultry | < = 1 per week | 85.2 | 92.3 | 93.4 | 99.0 | 54.7 | 40.3 | 70.7 | 89.6 | 45.2 |
| | 2–3 times per week | 14.5 | 7.3 | 4.8 | 1.0 | 31.6 | 30.0 | 26.7 | 8.9 | 20.6 |
| | 4 or more times per week | 0.3 | 0.3 | 1.7 | 0.0 | 13.7 | 29.7 | 2.6 | 1.4 | 34.2 |
| Legumes | < = 1 per week | 57.4 | 67.3 | 22.5 | 69.1 | 17.7 | 20.7 | 13.7 | 29.7 | 51.2 |
| | 2–3 times per week | 41.1 | 26.0 | 31.5 | 22.8 | 62.2 | 50.0 | 45.9 | 33.4 | 28.6 |
| | 4 or more times per week | 1.5 | 6.7 | 46.0 | 8.1 | 20.1 | 29.3 | 40.4 | 36.9 | 20.3 |
| Nuts and seeds | < = 1 per week | 26.9 | 63.0 | 89.6 | 72.2 | 32.4 | 31.4 | 49.2 | 77.8 | 27.9 |
| | 2–3 times per week | 64.5 | 24.7 | 7.6 | 15.8 | 57.4 | 44.8 | 21.5 | 19.3 | 30.9 |

*(Continued)*

**Table 3.** (Continued)

| | Frequency (%) | Burkina Nouna | Burkina Ouaga | Ethiopia Addis | Ethiopia Kersa | Nigeria Ibadan | Nigeria Lagos | Tanzania Dar es Salaam | Tanzania Dodoma | Ghana Kintampo |
|---|---|---|---|---|---|---|---|---|---|---|
| | 4 or more times per week | 8.6 | 12.3 | 2.8 | 12.1 | 10.2 | 23.8 | 29.3 | 2.9 | 41.2 |
| Dairy | < = 1 per week | 38.0 | 76.7 | 58.8 | 38.9 | 17.7 | 22.4 | 59.3 | 74.1 | 61.1 |
| | 2–3 times per week | 53.4 | 18.7 | 20.8 | 29.2 | 60.6 | 43.8 | 21.5 | 15.3 | 18.9 |
| | 4 or more times per week | 8.6 | 4.7 | 20.4 | 31.9 | 21.7 | 33.8 | 19.2 | 10.7 | 19.9 |
| Eggs | < = 1 per week | 68.2 | 73.7 | 73.7 | 74.5 | 16.6 | 19.0 | 73.0 | 87.6 | 52.8 |
| | 2–3 times per week | 27.2 | 23.7 | 19.4 | 22.5 | 49.9 | 34.1 | 18.6 | 9.5 | 24.3 |
| | 4 or more times per week | 4.6 | 2.7 | 6.9 | 3.0 | 33.5 | 46.9 | 8.5 | 2.9 | 22.9 |
| Whole grains | < = 1 per week | 16.4 | 41.3 | 66.4 | 12.4 | 26.8 | 35.5 | 44.0 | 34.6 | 31.9 |
| | 2–3 times per week | 41.3 | 37.0 | 19.4 | 35.9 | 53.6 | 49.7 | 28.3 | 10.4 | 22.3 |
| | 4 or more times per week | 42.3 | 21.7 | 14.2 | 51.7 | 19.6 | 14.8 | 27.7 | 55.0 | 45.9 |
| Liquid vegetable oils | < = 1 per week | 16.4 | 42.0 | 19.0 | 17.5 | 4.6 | 9.3 | 57.0 | 9.8 | 31.2 |
| | 2–3 times per week | 56.2 | 28.7 | 0.7 | 15.8 | 23.1 | 22.8 | 7.8 | 25.7 | 31.6 |
| | 4 or more times per week | 27.5 | 29.3 | 80.3 | 66.8 | 72.4 | 67.9 | 35.2 | 64.6 | 37.2 |

* Ouga—Ouagadougou; Burkina—Burkina Faso; Addis–Addis Ababa.

of Dodoma (55%), Kersa (52%), Kintampo (46%) and Nouna (42%). Consumption of nuts and seeds was low, Dairy intake was also low with at least 58% of the respondents reporting very low consumption in Ouagadougou, Addis Ababa, Kintampo and Tanzania. Egg consumption was low in Tanzania, Burkina Faso and Ethiopia with 68% of the respondents reporting consumption less than twice in the previous week.

**Consumption of unhealthy foods.** Table 4 describes the frequency of consumption of unhealthy foods. In Kintampo (36%), Ibadan (29%) and Lagos (27%) respondents reported higher red meat consumption (four or more times a week). Respondents in urban sites (Addis Ababa 22%, Lagos 22% and Dar es Salaam 18%) had relatively higher consumption of sugar-sweetened beverages, compared to rural sites. Processed meat was not commonly consumed, and sweets intake was low across all sites but slightly higher in Nigeria and Dar es Salaam. Finally, intake of refined grains was frequent across all study sites, and the majority of respondents in Kintampo (67%) reported consuming potatoes, roots and tubers four or more times a week.

**Changes in overall diet quality.** Fig 3 shows changes in the frequency of consumption of PDQS food groups comparing the time before the COVID-19 pandemic to the first (July to November 2020) and second (July to December 2021) round surveys. The frequency of consumption of both healthy and unhealthy PDQS food groups was lower in Round 1 and Round 2 surveys (during the COVID-19 pandemic), compared to recalled consumption prior to the pandemic (except for SSB). In general, consumption declined most during the Round 1 survey compared with the period before COVID-19 and started to improve in Round 2. Data were

**Table 4. Frequency of consumption of unhealthy PDQS food groups in Burkina Faso, Ethiopia, Nigeria, Tanzania and Ghana in the Round 2 study.**

| | Frequency (%) | Burkina Nouna | Burkina Ouaga | Ethiopia Addis | Ethiopia Kersa | Nigeria Ibadan | Nigeria Lagos | Tanzania Dar es Salaam | Tanzania Dodoma | Ghana Kintampo |
|---|---|---|---|---|---|---|---|---|---|---|
| Red meat | 4 or more times per week | 6.2 | 6.3 | 11.8 | 0.0 | 29.0 | 27.3 | 6.8 | 8.9 | 35.6 |
| | 2–3 times per week | 59.6 | 24.3 | 25.6 | 0.7 | 44.2 | 46.2 | 34.9 | 26.5 | 23.3 |
| | < = 1 per week | 34.5 | 69.3 | 62.6 | 99.3 | 26.8 | 26.6 | 58.3 | 64.6 | 41.2 |
| Processed meats | 4 or more times per week | 0.0 | 0.3 | 0.0 | 0.0 | 3.8 | 6.9 | 0.3 | 0.0 | 4.0 |
| | 2–3 times per week | 5.9 | 0.3 | 0.0 | 0.0 | 27.4 | 29.0 | 4.2 | 0.0 | 9.6 |
| | < = 1 per week | 94.1 | 99.3 | 100.0 | 100.0 | 68.9 | 64.1 | 95.4 | 100.0 | 86.4 |
| Refined grains | 4 or more times per week | 20.7 | 30.7 | 39.1 | 46.0 | 53.4 | 54.8 | 52.8 | 31.4 | 59.8 |
| | 2–3 times per week | 49.1 | 42.0 | 29.8 | 41.3 | 38.9 | 36.2 | 28.3 | 21.9 | 21.9 |
| | < = 1 per week | 30.3 | 27.3 | 31.1 | 12.8 | 7.8 | 9.0 | 18.9 | 46.7 | 18.3 |
| SSBs | 4 or more times per week | 2.8 | 2.0 | 21.5 | 2.4 | 16.9 | 22.4 | 18.2 | 5.2 | 12.6 |
| | 2–3 times per week | 42.0 | 20.3 | 13.8 | 22.2 | 46.1 | 36.2 | 29.6 | 12.7 | 21.9 |
| | < = 1 per week | 55.3 | 77.7 | 64.7 | 75.5 | 37.0 | 41.4 | 52.1 | 82.1 | 65.5 |
| Sweets and ice cream | 4 or more times per week | 8.6 | 5.0 | 3.5 | 3.4 | 12.1 | 12.4 | 3.3 | 14.1 | 6.3 |
| | 2–3 times per week | 38.6 | 16.3 | 6.2 | 11.4 | 41.8 | 32.1 | 35.8 | 23.1 | 11.6 |
| | < = 1 per week | 52.8 | 78.7 | 90.3 | 85.2 | 46.1 | 55.5 | 60.9 | 62.8 | 82.1 |
| Potatoes, roots and tubers | 4 or more times per week | 1.9 | 0.3 | 10.7 | 18.8 | 23.9 | 29.0 | 37.1 | 14.4 | 66.5 |
| | 2–3 times per week | 29.6 | 14.0 | 35.6 | 14.4 | 60.3 | 46.2 | 34.9 | 43.8 | 17.3 |
| | < = 1 per week | 68.5 | 85.7 | 53.6 | 66.8 | 15.8 | 24.8 | 28.0 | 41.8 | 16.3 |

*SSBs—Sugar Sweetened Beverages; Ouaga—Ouagadougou; Burkina—Burkina Faso; Addis–Addis Ababa.

captured around the same months for both surveys, which may decrease the influence of seasonality. However, there were exceptions. The frequency of consumption of dark green vegetables declined by 1 day and liquid vegetable oil consumption by 0.5 days per week on average in Round 2 compared to Round 1. Other food groups for which consumption was lower in Round 2 include poultry, nuts, and seeds.

*Overall diet quality*. The median PDQS (IQR) was 19 [17, 22], and PDQS was highest in Nigerian sites and lowest in Ouagadougou, Addis Ababa, Kersa and Dodoma. **Fig 4** shows mean PDQS before COVID-19, early during the time of COVID-19 in 2020 (Round 1), and later during the COVID-19 pandemic in 2021 (Round 2). Overall, PDQS had returned to pre-COVID-19 levels across sites by Round 2 of the survey but remained low. There are, however, site-specific differences. In Nouna and Addis Ababa, PDQS was slightly higher in Round 2 of the survey compared to previous times. In all other sites, PDQS was lower in Round 2, compared to before the pandemic. Tanzania and Ghana did not participate in round 1 data collection.

For the DDS food groups, the majority of respondents reported decreased consumption, with notable declines for dark green vegetables, other vegetables, staples, meats, beans and

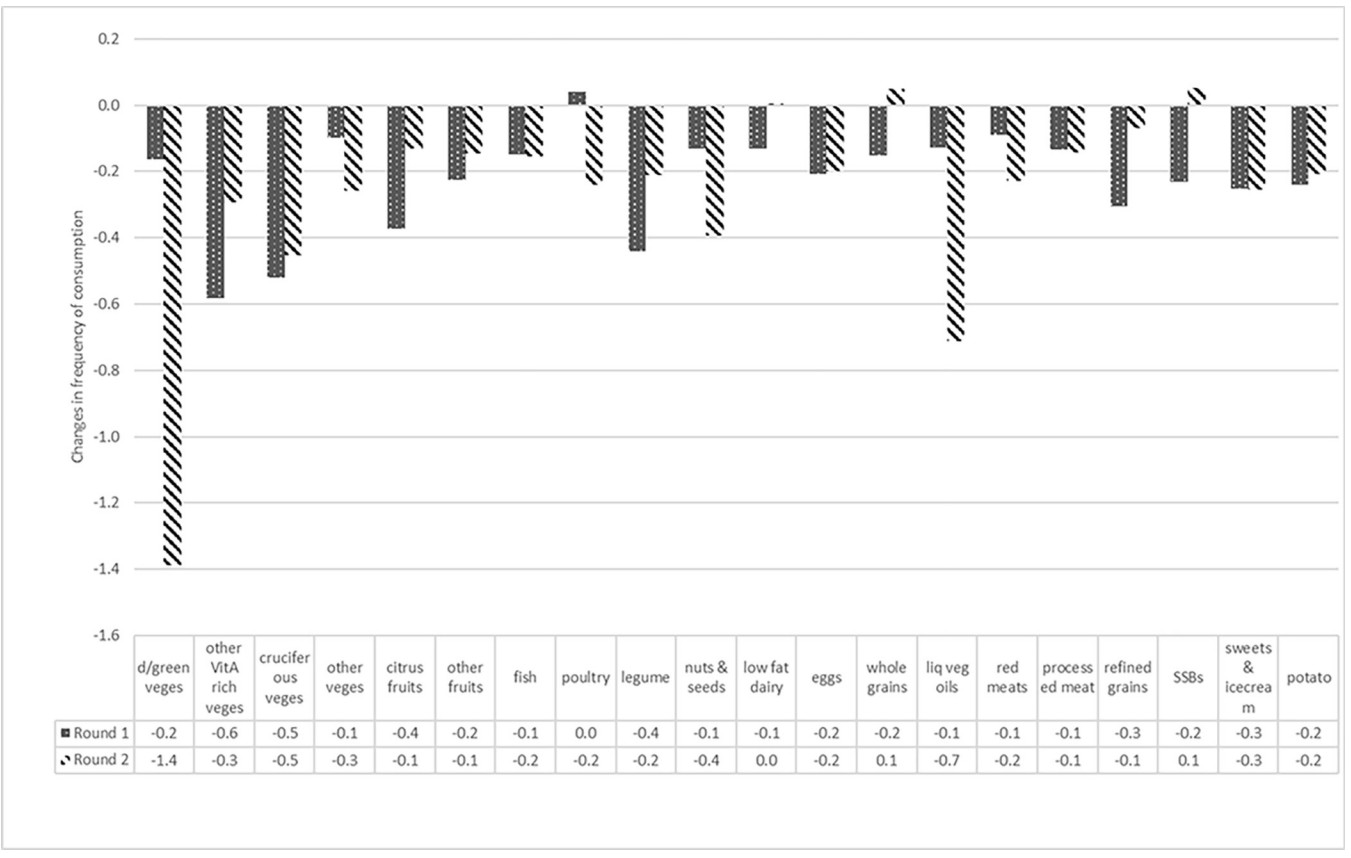

**Fig 3. Changes in frequency of consumption of PDQS food groups comparing the time before the time of COVID-19 to the first and second round surveys.**

peas and fruits (S1 Fig). Further, the DDS for all sites at the different survey rounds are shown in S2 Fig. Overall DDS was lower in the first survey in 2020 and improved at the time of the second survey in 2021 in most sites, to levels closest to pre-COVID estimates. However, in Burkina Faso sites, in Kersa (Ethiopia) and Ibadan (Nigeria) DDS was lower in the second round compared to pre-COVID times (S2 Fig).

Table 5 shows the factors associated with diet quality across the five countries in Round 2 of the ARISE study. In a multivariate analysis, respondents residing in Nigeria (estimate: 2.84, 95% CI: 2.26,3.41) and Ghana (estimate: 0.93, 95% CI: 0.37,1.49) had higher PDQS compared to those residing in Burkina Faso. Respondents aged 30–39 years (estimate: 0.77, 95% CI: 0.35,1.19) and those 40 years or older (estimate: 0.72, 95% CI: 0.30,1.13) had higher PDQS compared to those aged 29 years or younger. Male respondents (estimate: -0.54, 95% CI: -0.88, -0.20) and those with no education had poorer diet quality (estimate: -0.40, 95% CI: -0.76, -0.03), and secondary school or higher education was associated with a higher PDQS (estimate: 0.73, 95% CI: 0.32,1.15) compared to having primary school education. Respondents who identified as Catholic (estimate: 0.66, 95% CI: 0.12,1.19) or Muslim (estimate: 0.51, 95% CI: 0.10,0.93) reported higher PDQS compared to Orthodox Christian respondents. Farmers and those engaged in casual labor reported lower PDQS (estimate: -0.60, 95% CI: -1.11, -0.09) compared to those that were formally employed.

Unemployed respondents had on average lower diet quality (estimate: -0.38, 95% CI: -0.69, -0.06) compared to those whose employment status was unchanged during the COVID-19

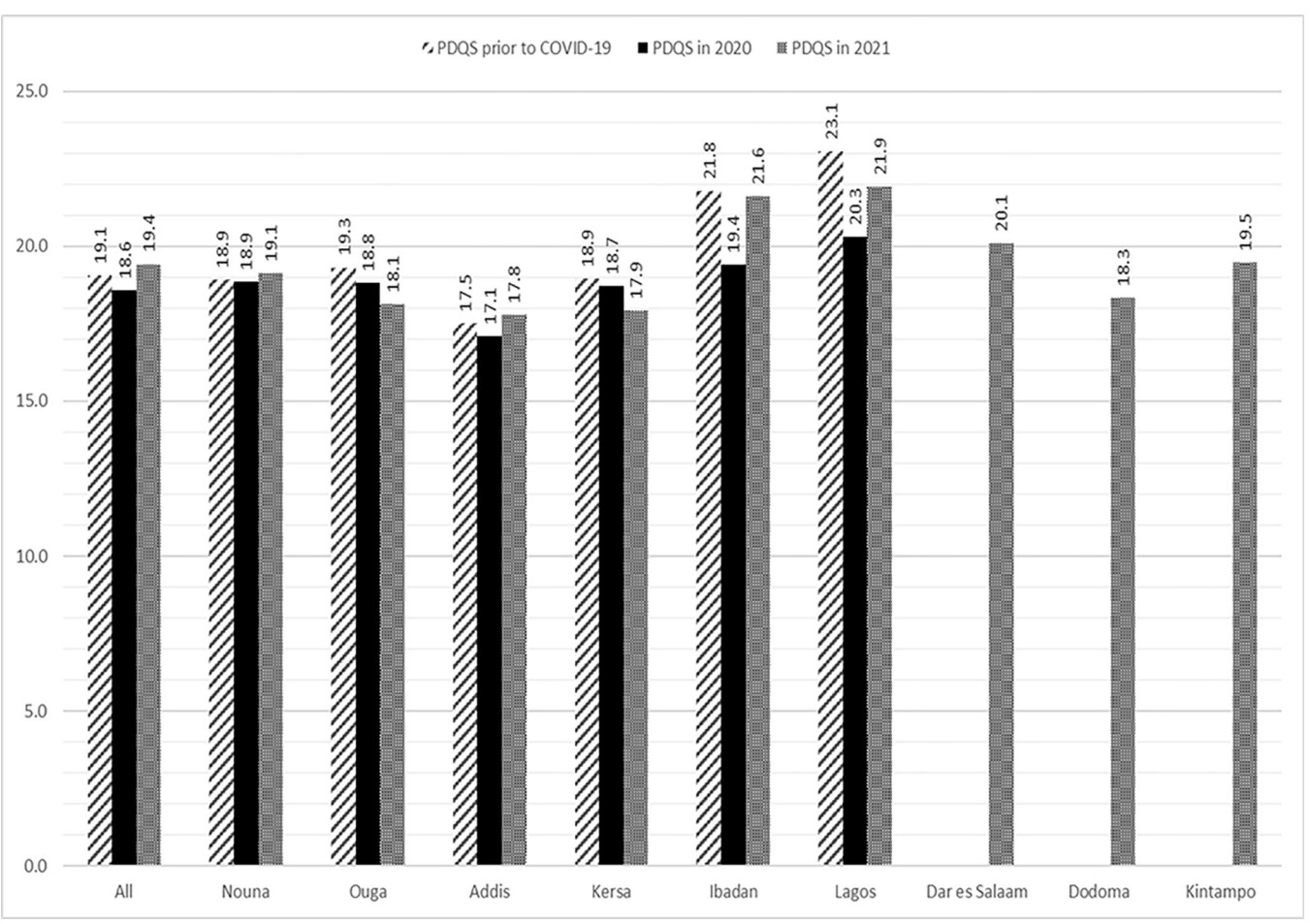

**Fig 4. Mean Prime Diet Quality Scores (PDQS) before, early and later in the COVID-19 pandemic across five countries.**

pandemic. Those reporting going for an entire day without eating had lower PDQS than those who did not (estimate: -1.10, 95% CI: -1.53, -0.67). Lower crop production (estimate: -0.87, 95% CI: -1.28, -0.46) or not engaging in farming (estimate: -1.38, 95% CI: -1.74, -1.02) were associated with lower PDQS compared to having unchanged levels of crop production. Being in the middle tertile of the wealth index (estimate: 0.48, 95% CI: 0.14,0.81) was associated with higher PDQS compared to being in the highest tertile.

In sensitivity analysis controlling for PDQS before the COVID-19 pandemic and restricted to the three countries with Round 1 data, PDQS before the COVID-19 pandemic was positively associated with PDQS later in the Round 2 survey (estimate 0.09, 95% CI: 0.03,0.14) after adjusting for other factors. We found country-level differences in PDQS, with respondents in Nigeria having higher PDQS (estimate 1.93, 95% CI: 0.82,3.04) and those in Ethiopia having lower PDQS (estimate: -0.85, 95% CI: -1.54, -0.15) compared to those residing in Burkina Faso. Higher fruit prices were associated with lower PDQS (estimate: -1.43, 95% CI: -2.33, -0.52). Secondary school education or higher was associated with higher diet quality (estimate: 1.25, 95% CI: 0.47,2.03) compared to primary school education, as was being self-employed or a student (estimates: 0.98, 95% CI:0.27,1.70) compared to being employed. Lower or reduced salary (estimate -0.85, 95% CI: -1.62, -0.08), lower crop production (estimate: -1.04, 95% CI: -1.68, -0.41) and not farming (estimate -1.23, 95% CI: -1.84, -0.62) were associated with lower diet quality after adjusting for other factors.

**Table 5. Factors associated with PDQS in Round 2 of the ARISE study across five countries.**

| | Univariate | Multivariate model[a] | Multivariate secondary analysis[b] |
|---|---|---|---|
| | Estimate (95% CI) | Estimate (95% CI) | Estimate (95% CI) |
| PDQS before COVID-19 | 0.16(0.11,0.22)*** | - | 0.09(0.03,0.14)* |
| Country | | | |
| Burkina Faso | ref | Ref | ref |
| Ethiopia | -0.78(-1.17,-0.39)*** | -0.37(-0.86,0.11) | -0.85(-1.54,-0.15)* |
| Nigeria | 3.11(2.72,3.49)*** | 2.84(2.26,3.41)*** | 1.93(0.82,3.04)** |
| Tanzania | 0.53 (0.14,0.91)* | -0.13(-0.74,0.49) | - |
| Ghana | 0.82(0.34,1.30)** | 0.93(0.37,1.49)** | - |
| Location | | | |
| Rural area | -0.11(-0.39,0.11) | -0.13(-0.51,0.24) | 0.11(-0.66,0.88) |
| Urban | ref | ref | ref |
| Staple prices | | | |
| No change or decreased | ref | ref | ref |
| Increased | 0.32(-0.01,0.65) | -0.01(-0.63,0.62) | 0.05(-1.28,1.37) |
| Pulse prices | | | |
| No change or decreased | ref | ref | ref |
| Increased | 0.32(-0.01,0.66) | 0.21(-0.37,0.79) | 0.78(-0.37,1.94) |
| Fruits prices | | | |
| No change or decreased | ref | ref | ref |
| Increased | 0.27(-0.04,0.58) | -0.45(-0.98,0.08) | -1.43(-2.33,-0.52) ** |
| Vegetable prices | | | |
| No change or decreased | ref | ref | ref |
| Increased | 0.27(-0.03,0.60) | -0.15(-0.71,0.41) | 0.38(-0.64,1.39) |
| Animal-source foods prices | | | |
| No change or decreased | ref | ref | ref |
| Increased | 0.49(0.13,0.84)* | 0.41(-0.10,0.92) | -0.31(-1.42,0.81) |
| Age (years) | | | |
| 20–29 | ref | ref | ref |
| 30–39 | 0.56(0.11,1.01)* | 0.77(0.35,1.19)*** | 0.35(-0.53,1.22) |
| ≥ 40 | 0.43(0.02,0.82)* | 0.72(0.30,1.13)** | -0.04(-0.91,0.84) |
| Sex | | | |
| Female | ref | ref | ref |
| Male | -0.36(-0.64,-0.08)* | -0.54(-0.88,-0.20)** | 0.30(-0.36,0.96) |
| Education | | | |
| None or incomplete primary | -0.65(-0.99,-0.33)*** | -0.40(-0.76,-0.03)* | -0.17(-0.77,0.43) |
| Primary school or incomplete secondary | ref | ref | ref |
| Secondary school or higher | 1.72(1.37,2.06)*** | 0.73(0.32,1.15)** | 1.25(0.47,2.03)* |
| Household head | -0.64(-0.93,-0.35)*** | 0.12(-0.24,0.47) | 0.08(-0.58,0.74) |
| Household size | -0.06(-0.10,-0.02)** | 0.02(-0.01,0.07) | 0.01(-0.04,0.07) |
| Religion | | | |
| None | 0.01(-1.18,1.21) | -0.48(-0.64,1.61) | -2.52(-5.23,0.20) |
| Catholic | -0.56(-1.05,-0.08)* | 0.66(0.12,1.19)* | -0.79(-1.80,0.21) |
| Muslim | -0.89(-1.25,-0.54)*** | 0.51(0.10,0.93)* | -0.49(-1.30,0.33) |
| Orthodox Christian | ref | ref | ref |
| Protestant or other | -0.84(-1.26,-0.42)*** | 0.27(-0.25,0.78) | -0.59(-1.67,0.48) |
| Occupation | | | |
| Unemployed | -2.20(-2.69,-1.70)*** | -0.39(-0.93,0.16) | 0.21(-0.69,1.12) |

*(Continued)*

**Table 5.** (Continued)

|  | Univariate | Multivariate model[a] | Multivariate secondary analysis[b] |
|---|---|---|---|
|  | Estimate (95% CI) | Estimate (95% CI) | Estimate (95% CI) |
| Farmer or casual labor | -2.06(-2.43,-1.70)*** | -0.60(-1.11,-0.09)* | 0.52(-0.35,1.39) |
| Employed | ref | ref | ref |
| Student, self-employed or other | -0.65(-1.03,-0.27)** | 0.01(-0.37,0.40) | 0.98(0.27,1.70)* |
| Effects of COVID-19 |  |  |  |
| None, unemployed | -0.94(-1.25,-0.62)*** | -0.38(-0.69,-0.06)* | -0.33(-0.90,0.23) |
| None, no change in employment status | ref | ref | ref |
| Lost employment | -0.01(-0.55,0.57) | 0.44(-0.09,0.98) | -0.31(-0.48,1.11) |
| Changed occupation | 0.18(-0.64,1.00) | 0.12(-0.64,0.88) | 0.20(-1.35,1.76) |
| Other (specify) | 0.82(0.23,1.42)* | 0.42(-0.15,1.00) | -0.58(-1.94,0.79) |
| COVID-19 Impact on income |  |  |  |
| Income is unchanged | ref | ref | ref |
| Lost or reduced salary from employer | 0.86(0.40,1.32)** | -0.19(-0.67,0.30) | -0.85(-1.62,-0.08)* |
| Lost or reduced income farming business | 0.37(0.06,0.68) | -0.13(-0.45,0.19) | -0.51(-1.07,0.05) |
| Increased salary from employer or farming business | 0.03(-1.11,1.17) | -0.36(-1.40,0.69) | -0.39(-2.16,1.37) |
| **Food security** |  |  |  |
| Went without eating for a whole day (past month) |  |  |  |
| No | ref | ref | ref |
| Yes | -0.71(-1.17,-0.24)** | -1.10(-1.53,-0.67)*** | -1.28(-2.03,-0.52)** |
| Own crop production affected |  |  |  |
| Unchanged | ref | ref | ref |
| Production has decreased | -1.06(-1.46,-0.67)*** | -0.87(-1.28,-0.46)*** | -1.04(-1.68,-0.41)** |
| Production has increased | 1.16(0.10,2.22)* | 0.18(-0.81,1.17) | 0.58(-1.13,2.29) |
| Does not farm | -0.22(-0.53,0.09) | -1.38(-1.74,-1.02)*** | -1.23(-1.84,-0.62)*** |
| Wealth index |  |  |  |
| Wealth tertile 1 | 0.02(-0.31,0.35) | -0.21(-0.51,0.10) | 0.14(-0.39,2.26) |
| Wealth tertile 2 | 1.07(0.71,1.42)*** | 0.48(0.14,0.81)* | 0.43(-0.16,1.02) |
| Wealth tertile 3 | ref | ref | ref |

Acronyms: PDQS: Prime Diet Quality Score

\* <0.05

\*\*<0.01

\*\*\*<0.001

a/Based on Round 2 data only

b/ Model restricted to individuals with round 1 and 2 data (N = 929) and adjusted for PDQS score prior to the COVID-19 pandemic (measured at Round 1).

## Discussion

We evaluated the impacts of COVID-19 on factors known to influence food consumption such as food prices, food security and crop production, and how these and other factors affected diet quality two years into the COVID-19 pandemic in Burkina Faso, Ethiopia, Ghana, Nigeria and Tanzania. We found that food prices remained higher than expected during the pandemic, consumption of healthy food groups remained low, and that diet quality showed signs of slight recovery since 2020 but remained poor. Food insecurity, low agriculture production, and some occupations were negatively associated with PDQS, while higher socioeconomic status was positively associated with diet quality.

In this study, we found evidence of the continued impact of COVID-19 on food prices, with the majority of respondents indicating that food prices were higher than before the

pandemic. In a study conducted across several countries early in the COVID-19 emergency in June 2020, there was evidence that the prices of maize, sorghum, imported rice and rice in SSA were higher than expected [29]. Price increases were attributed to movement restrictions and lockdowns, with economic factors such as exchange rate, and inflation also contributing [29]. Other studies indicated that a global slowdown due to COVID-19 and its related impact on Gross Domestic Product (GDP) could lead to impacts on food security and consumption, through increased unemployment, reduced trade and decreased production [30].

In this study, we found that lower agricultural production, not participating in crop production, as well as food insecurity, were associated with the consumption of lower-quality diets. The impacts of COVID-19 on agricultural production and food security have also been documented in previous studies. In one study, the COVID-19 pandemic was shown to impact bean production by small-scale farmers in SSA, with disruptions in access to seeds, inputs, and labor, among other things [31]. Lower agricultural production can impact the consumption of quality diets by decreasing the availability of foods and increasing prices for nutritious foods. In Nigeria, disruptions due to COVID-19 restrictions early in the pandemic included lockdowns, curfews and travel restrictions in urban areas and these affected the ability of local authorities to support agricultural production, disrupted trade and decreased access to healthy diets [32]. Although restrictions across countries have been lifted, the lasting impact may be due to impacts on regional and international food trade that have affected food prices [32].

We found that diet quality and diversity scores showed initial dips and later showed small increases later during the COVID-19 crisis; sometimes returning closer to pre-COVID states. However, despite these increments, it was still clear that consumption of key micronutrient-rich food groups such as dark green vegetables, other vegetables, poultry, nuts and seeds, and liquid vegetable oils had not recovered to pre-COVID-19 levels. Similar observations were noted for red meat consumption. Further, the noted recovery of diet quality scores is small and both diet quality and diversity remain low across all the countries assessed. A study in Mexico also investigated the factors associated with diet quality early and later during the COVID-19 pandemic and found that the consumption of healthy foods and food security declined, however, consumption of unhealthy foods increased [33]. In another study in the United States, individuals reporting food insecurity also had lower consumption of both healthy and unhealthy foods [34]. In our study, there were country-specific differences with greater recovery in diet quality experienced in sites in Nigeria and Addis Ababa. In contrast, other sites continued to experience declining diet quality and diversity. These differences could reflect the differences in the states' capacity to mitigate against the COVID-19 impacts and the availability of social protection. In this study, however, the availability of social protection to address the impacts of COVID-19 was very limited.

The impacts of COVID-19 on the African continent, particularly on food security and diets, are expected to be dire, given that purchasing power and social safety nets are already limited [35]. In several countries, the impacts of COVID-19 on food prices and quality have been due to decreased economic activity, lower income and declines in employment and income. We found that loss of employment was infrequently reported. However, nearly 50% of the respondents indicated that they had lost income from employment, farming or business activities. The impacts of the loss of income from employment and income-generating activities on diets can be through loss of purchasing power [12]. The effects of COVID-19 related income shocks on food security and diets have been reported in Uganda and Kenya, with poor households most affected [36, 37]. For example, loss of income during COVID-19 was reported by farmers in Burundi (36%), Uganda (20%), and Kenya (3%). In our study, loss of income from farming and business was greatest in Kintampo, Dar es Salaam and Ibadan, with rural areas more likely to experience a loss due to impacts on agriculture. Previous studies

have indicated rural areas and farming communities might be more resilient to the impacts of COVID-19 due to shorter value chains and reliance by smallholder farmers on their production [12]. In our study, the impacts on rural and urban locations depend on the country context; for example, the largest declines in DDS were observed in urban sites such as Ouagadougou and rural sites of Kersa and Nouna.

We found that the older or more educated respondents with higher socioeconomic status had higher diet quality in the study. We also found that those engaged in farming or casual labor in our study tended to have lower diet quality, reinforcing this observation. These individuals are less likely to have enough savings to cope with shocks such as the COVID-19 pandemic and may have been highly vulnerable even before the shock. Economic stimulus packages and investment in sectors such as agriculture have been proposed as possible solutions to address the challenges posed by the COVID-19 pandemic in Africa and to promote recovery [31]. However, these were not readily available to communities and respondents in our study settings.

Overall, our study findings are that during the COVID-19 pandemic respondents reported food prices increased and lower diet quality diets. Those living in urban areas were affected more compared to rural communities in terms of price increases. In rural areas more participants engaged in agricultural production and if production was maintained, the COVID-19 effects were less likely to be reported. Those living in rural areas may be protected from market disruptions. However, those that reported lower agriculture production or no agriculture production had poorer diets. Furthermore, COVID-19 related disruptions differed among the countries and locations surveyed, indicating an important role of prevailing local conditions prior to the pandemic.

Moreover, those who were already vulnerable, for example with low education, farmers and causal laborers were more likely to have poorer diets during the pandemic. Studies have suggested that the greatest impact of COVID-19 in the African context has been felt by those who were already vulnerable such as those of low social economic status eg informal workers, those reliant on daily income or those facing social and economic vulnerability [12]. This was evident in our study.

Our findings are also consistent with the findings of a systematic review of the impact of COVID-19 on diet quality and nutrition in LMICs (including Ethiopia and Nigeria) in 2021 that found that impacts varied in intensity and duration and were informed by the stringency and duration of national COVID-19 shutdowns and extent of reliance on local compared to international food markets [12]. Thus urban areas which are more integrated with external markets may be more vulnerable to price fluctuations, Additionally, locations in rural areas and those with shorter value chains such as small-scale farming households were more resilient and were less adversely affected by COVID-19 related disruptions [12].

Our study has several strengths. First, we conducted a repeated cross-sectional study in 3 countries, allowing us to track changes in food security and diet intake at different times during the COVID-19 pandemic. Further, we collected data on the same individuals in some contexts. A limitation of the study is that we had a loss to follow-up in round 2 of the survey, limiting our ability to look at changes over time prospectively for more households. Loss to follow-up is, however, expected in telephone surveys, especially in our study contexts. Also, there may be measurement errors as a result of self-reported dietary intake and food pricing. However, we do not think that measurement error will vary by the outcomes.

## Conclusion

Our study finds that households in study countries reported higher food prices, and lower diet quality during the COVID-19 pandemic. We found that effects differed by country and local

contexts, suggesting the need for a nuanced understanding of the influence of COVID-19 on diets. Those who were already vulnerable before the pandemic such as casual laborers and the unemployed were more likely to be affected, as well as those not engaged in farming and more reliant on markets, although this differed by country and rural and urban location. Those whose agriculture production was unaffected and likely to subsist on their own production may be protected from market disruptions due to the COVID-19 pandemic. Further, while there was some recovery in the quality of diets consumed as the COVID pandemic continued, recovery lagged behind for the consumption of some nutrient-rich foods. Dietary quality and diversity were already low prior to the pandemic and remain a cause for concern.

All these factors including higher than usual food prices and poor diet quality experienced in the study have health implications. Efforts should continue to improve diet quality through mitigation measures, including social protection for people for sustained nutrition recovery. More importantly, systematic efforts to improve the underlying causes of poor diet quality through transforming food system value chains through relevant programs and national policies are critical.

## Supporting information

**S1 Fig. Change in consumption of MDDW food groups prior to and in Round 2 of the COVID-19 study.**
(TIF)

**S2 Fig. Mean diet diversity scores (DDS) prior to, early during COVID-19 and later in the COVID-19 pandemic across 5 countries.**
(PUB)

## Acknowledgments

We thank all study participants and data collectors for contributing to this study. The survey team in Ghana is grateful for support from the Kintampo Health Research Centre of Ghana Health Service, and the community leadership of Kintampo North Municipality and Kintampo South District. We acknowledge institutional support from Harvard T.H. Chan School of Public Health, Boston, MA; Harvard University Center for African Studies, Boston, MA; Heidelberg Institute of Global Health, Germany and the George Washington University Milken Institute of Public Health, Washington, DC.

## Author Contributions

**Conceptualization:** Abbas Ismail, Isabel Madzorera, Edward A. Apraku, Amani Tinkasimile, Dielbeogo Dasmane, Millogo Ourohire, Nega Assefa, Angela Chukwu, Firehiwot Workneh, Elena Hemler, Dongqing Wang, Sulemana W. Abubakari, Kwaku P. Asante, Till Baernighausen, Japhet Killewo, Ayoade Oduola, Ali Sie, Said Vuai, Emily Smith, Yemane Berhane, Wafaie W. Fawzi.

**Data curation:** Isabel Madzorera, Pascal Zabre, Frank Mapendo, Bruno Lankoande.

**Formal analysis:** Abbas Ismail, Isabel Madzorera.

**Funding acquisition:** Till Baernighausen, Emily Smith, Wafaie W. Fawzi.

**Methodology:** Abbas Ismail, Isabel Madzorera, Edward A. Apraku, Amani Tinkasimile, Dielbeogo Dasmane, Millogo Ourohire, Nega Assefa, Angela Chukwu, Firehiwot Workneh,

Dongqing Wang, Sulemana W. Abubakari, Till Baernighausen, Japhet Killewo, Ayoade Oduola, Abdramane Soura, Said Vuai, Emily Smith, Yemane Berhane, Wafaie W. Fawzi.

**Project administration:** Elena Hemler.

**Writing – original draft:** Abbas Ismail, Isabel Madzorera, Yemane Berhane.

**Writing – review & editing:** Abbas Ismail, Isabel Madzorera, Edward A. Apraku, Amani Tinkasimile, Dielbeogo Dasmane, Pascal Zabre, Millogo Ourohire, Nega Assefa, Angela Chukwu, Firehiwot Workneh, Frank Mapendo, Bruno Lankoande, Elena Hemler, Dongqing Wang, Sulemana W. Abubakari, Kwaku P. Asante, Till Baernighausen, Japhet Killewo, Ayoade Oduola, Ali Sie, Abdramane Soura, Said Vuai, Emily Smith, Wafaie W. Fawzi.

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
