## [Decision Letter · Decision Letter 0]

23 Jan 2023

PONE-D-22-33938The COVID-19 pandemic and its prolonged impacts on food prices, food consumption and diet quality in sub-Saharan AfricaPLOS ONE

Dear Dr. Madzorera,

Thank you for submitting your manuscript to PLOS ONE. After careful consideration, we feel that it has merit but does not fully meet PLOS ONE’s publication criteria as it currently stands. Therefore, we invite you to submit a revised version of the manuscript that addresses the points raised during the review process.

In addition to addressing reviewers'' comments, please address the following comments as well:The link in line 183 is not working. It is suggested to move this link in the reference list, write the accessed date, and cite the needed information accordingly.There is a complete absence of literature review section. Please add a literature review section.The results of the study should be compared and contrasted to other published studies.The authors are urged to use paired t-test/Wilcoxon test to compare their variables of interest before and after COVID pandemic. The mathematical equations of the estimated models should be clearly specified in the statistical analysis section.The conclusion is very brief and weak, and hence It should be improved.Please make sure the article meets PLOS ONE’s style including the referencing style.

We look forward to receiving your revised manuscript.

Kind regards,

Mohammed Al-Mahish

Academic Editor

PLOS ONE

Journal Requirements:

3. Thank you for stating the following in the Funding Source Section of your manuscript: 

"This work was supported by institutional support from Harvard T.H. Chan School of Public Health, Boston, MA; Harvard University Center for African Studies, Boston, MA; Heidelberg Institute of Global Health, Germany, and the George Washington University Milken Institute of Public Health, Washington, DC."

"This work was supported by institutional support from Harvard T.H. Chan School of Public Health, Boston, MA (WF); Harvard University Center for African Studies, Boston, MA (WF); Heidelberg Institute of Global Health, Germany (TB), and the George Washington University Milken Institute of Public Health, Washington, DC (ES). The funders have no role in study design, data collection and analysis, decision to publish, or preparation of the manuscript."

Reviewers' comments:

Reviewer's Responses to Questions

**Comments to the Author**

1. Is the manuscript technically sound, and do the data support the conclusions?

Reviewer #1: Yes

Reviewer #2: Yes

2. Has the statistical analysis been performed appropriately and rigorously? 

Reviewer #1: Yes

Reviewer #2: Yes

3. Have the authors made all data underlying the findings in their manuscript fully available?

Reviewer #1: No

Reviewer #2: Yes

4. Is the manuscript presented in an intelligible fashion and written in standard English?

Reviewer #1: Yes

Reviewer #2: Yes

5. Review Comments to the Author

Reviewer #1: • I think the manuscript focused more on the diet quality rather than food price, as stated in Line 196. Therefore, I suggest to change the title to:

“The COVID-19 pandemic and its impacts on diet quality and food prices in sub-Saharan Africa”

• Line 69: in abstract, Dietary diversity score (DDS), should be mentioned in the Method used

• Line 69-70 could be done as one sentence, no need for repetition (data collected)

• Line 76-84: there is no need to write the numbers here, as they are mentioned inside the text, additional study results could be stated here

• Introduction: was written systematically

• Line 122-125: Reference order would be 12-13-14, not 12-14-13

• Line 150-151: All nine sites from the five countries (Figure 1) were included in the second round of data collection. (No need for repetition)

• Line 160: Study design, is clearly described and detailed, still a sentence is needed to clearly stated the total sample size in each round as stated in line 273

• Line 195 and 217: It is mentioned that the diet diversity score (DDS) was computed, still the DDS results are not shown on the results and discussion sections

• Line 204-205: In addition to mention previous study using PDQS, it is recommended to mention some studies used the same dependent variables to compare results

• Line 242: head of household (yes or no) what does (Yes) indicates? More explanation is needed

• Line 246: Wealth index, more elaboration is needed of factor analysis

• Line 252: Why GEE was adopted as an analytical tool? A brief justification is needed, with model specification

• Line 282 Table 1: for easy follow up and understanding, it is better to write only the percentage not numbers inside the Table

• Line 299-303 and line 239: These questions (worried about food, skipped a meal, gone for an entire day without eating.) are related to food consumption behaviors which is one of the types of food security measurement. Therefore, it is better to rephrase the sentences to reflect that. The same comment is applicable to Table 2

• Line 314: Table 2:

- There is no need to repeat “Impact of COVID-19” on the subheading as it was reflected of Table title.

- Table title it is better to write five rather than 5.

- Is there any significance difference between these percentages???? may be another test is required

• Line 324: All figures numbers should be revised to match the figures attached at the end of the manuscript

• Line 335: It is better to shorten the Fig 2 title

• Lines 340 and 360: Also applicable to line 373

- “consumption of healthy PDQS food groups”, is it a right scientific terminology????

- Unit time is required with PDQS (week or 24 hrs)

• Line 378-380: What is the source for data for” consumption of food groups at the time before the COVID-19 pandemic”, it should be stated in material and method section

• Line 608: “2020;4:1401-.” Please delete the dash (-) at the end

• Line 637: The publication year is missing

Reviewer #2: The paper addresses an important recent topic and covered sub-researched countries in Africa. The writing is good and easy to follow; however, there is a need to improve some parts as discussed below. It would be also helpful if page number has been added to the document.

Introduction: The introduction should clearly state the main contributions of the study to strengthen the argument and ensure the importance of this work related to other relevant studies. A clearly stated and well-defined contribution is the merit of any research. Authors only indicated “Therefore, it is important that as the COVID-19 pandemic persists we continue to assess its impact on the nutrition and health of the households in SSA (12).” Then, the authors have details the strengths of the work on page 39 in the discussion. These strengths /contributions should also be indicated in the introduction.

Please add to the study design justification that the differences between the two samples/rounds will not cause issues in comparisons (across countries, across rounds if applicable) in the results section, especially that you conducted country-level differences in PDQS.

Conclusion should include a summary of what the study is about, not only the outcomes.

Minor comments are as follows:

Table font is different than the text font and style.

Texts in tables and figures should be in black to be more readable, it is in gray now.

Page 22: “Loss of or reduced salaries” I think you meant “lost or reduced salaries”

Page 23 and 24: Not sure what do “(not shown) and “(results not shown)” mean? If you meant the results are not shown in the Table, please state that clearly.

6. PLOS authors have the option to publish the peer review history of their article (what does this mean?). If published, this will include your full peer review and any attached files.

Reviewer #1: No

Reviewer #2: No

---

## [Author Response · Author response to Decision Letter 0]

17 May 2023

PONE-D-22-33938

The COVID-19 pandemic and its prolonged impacts on food prices, food consumption and diet quality in sub-Saharan Africa

PLOS ONE

Dear Dr. Madzorera,

Comment: Thank you for submitting your manuscript to PLOS ONE. After careful consideration, we feel that it has merit but does not fully meet PLOS ONE’s publication criteria as it currently stands. Therefore, we invite you to submit a revised version of the manuscript that addresses the points raised during the review process.

Response: We would like to thank the Editor and Reviewers for your review of the manuscript and feedback provided. We have found the review process and insights provided helpful and it has helped us improve the quality of our manuscript. We think our audience will certainly benefit from the improved paper. 

Comment: In addition to addressing reviewers'' comments, please address the following comments as well:

1. The link in line 183 is not working. It is suggested to move this link in the reference list, write the accessed date, and cite the needed information accordingly.

Response: We thank the Editor for this comment. As suggested, we have replaced the link with a reference and in the manuscript. We have updated the text as follows,

“The design and methods of the round 2 survey are detailed on the Harvard University Center for African Studies website (1).” (Lines 696-697, page 10)

Comment: 

2. There is a complete absence of literature review section. Please add a literature review section.

Response: We acknowledge the Editor’s comments and agree that this is important. To address this concern, we have added additional literature to the manuscript, mainly in the discussion section. The changes are tracked in the resubmitted manuscript. Because there was limited literature on the topic prior to and during our study, we are limited in the literature we can add to the introduction section.

Comment:

3. The results of the study should be compared and contrasted to other published studies.

Response: We thank the reviewers for this comment. Comparisons to other studies were provided where available in the discussion section. It is important to note that unlike in other regions, studies on this topic have been limited in the African context and the literature does not cover in detail all the topics presented in the paper. Therefore, our review of literature is limited to studies available at the time. We have also added a few additional reports that we could find on the topic in the discussion section.

Comment:

4. The authors are urged to use paired t-test/Wilcoxon test to compare their variables of interest before and after COVID pandemic. 

Response: We thank the Reviewers for this suggestion. We have carefully considered this suggestion and do agree that what the authors suggest is an interesting research question. However, because of our study design and the nature of data collected, we believe it may not be advisable to apply the suggested method. We asked respondents to recall their usual consumption of foods prior to the COVID-19 pandemic and in the first and second round (or before and during COVID). With the suggested approach we would be evaluating differences in consumption prior to COVID-19 and later during the pandemic without considering possible confounding. However, given that we rely on self-reported/recalled data ( with a longer recall period for pre-COVID 19 diets), we worry about recall bias and confounding. This analysis proposed is possible if we wanted to restrict respondents who appeared in both rounds of data collection, calculate PDQS as a continuous variable, and compare between rounds using paired t-tests. However, we believe that the secondary analysis in table 5 is even better because we as we controlled for other variables. We believe this sufficiently addresses the concerns raised.

Comment:

5. The mathematical equations of the estimated models should be clearly specified in the statistical analysis section.

Response: We thank the reviewer for this comment. In the field of epidemiology, it is not the norm to include the mathematical models in the manuscript itself. Our target audience would not be expecting this. Nevertheless, we present the model for the main analysis shown in table 5 below.

YMean PDQS=�+�1X1+�2X2+�3X3+�4X4+�5X5+�nXn+

Comment:

6. The conclusion is very brief and weak, and hence It should be improved.

Response: We have noted this feedback. We have strengthened the conclusion as follows,

Overall, our study reveals that many on the African continent experienced higher foods prices, and lower diet quality during the COVID-19 pandemic. We found that effects differed by country and local contexts, suggesting the need for nuanced understanding of the influence of COVID-19. Further it was evident that those who were already vulnerable before the pandemic were more likely to be affected, as well as those not engaged in farming and more reliant on markets, although this differed by country and rural and urban location. Those whose agriculture production was unaffected and likely to subsist on their own production may be protected from market disruptions due to the COVID-19 pandemic. We found that while there was some recovery for diet quality, recovery lagged behind for consumption of some nutrient rich foods and that dietary diversity and quality were already low prior to the pandemic and remain a cause for concern. All these factors including higher than usual food prices and poor diet quality experienced in the study have health implications. Efforts should continue to improve diet quality through mitigation measures, including social protection for people for sustained nutrition recovery. Further, more systematic efforts to improve the underlying causes of poor diet quality through transforming food systems value chains through relevant programs and national policies is critical. (lines 1120-1141, pages 32-33)

Comment:

7. Please make sure the article meets PLOS ONE’s style including the referencing style.

Response: We thank the Editor for this additional guidance. We are submitting the documents required and will ensure that all documents adhere to the provided guidelines.

We look forward to receiving your revised manuscript.

Kind regards,

Mohammed Al-Mahish

Academic Editor

PLOS ONE

Journal Requirements:

Comment:

Response: We thank the reviewer for this feedback. We have formatted the manuscript using the guidelines provided.

Comment:

Response: We thank the reviewer. This comment has been addressed above. We have removed all information on funding sources from the manuscript as advised. We consent to the use of the following statement as our Funding statement:

"This work was supported by institutional support from Harvard T.H. Chan School of Public Health, Boston, MA (WF); Harvard University Center for African Studies, Boston, MA (WF); Heidelberg Institute of Global Health, Germany (TB), and the George Washington University Milken Institute of Public Health, Washington, DC (ES). The funders have no role in study design, data collection and analysis, decision to publish, or preparation of the manuscript."

Comment:

3. Thank you for stating the following in the Funding Source Section of your manuscript: 

"This work was supported by institutional support from Harvard T.H. Chan School of Public Health, Boston, MA; Harvard University Center for African Studies, Boston, MA; Heidelberg Institute of Global Health, Germany, and the George Washington University Milken Institute of Public Health, Washington, DC."

"This work was supported by institutional support from Harvard T.H. Chan School of Public Health, Boston, MA (WF); Harvard University Center for African Studies, Boston, MA (WF); Heidelberg Institute of Global Health, Germany (TB), and the George Washington University Milken Institute of Public Health, Washington, DC (ES). The funders have no role in study design, data collection and analysis, decision to publish, or preparation of the manuscript."

Response: We thank the Reviewers for this feedback. Please see response above. Our funding statement has been adjusted as follows:

"This work was supported by institutional support from Harvard T.H. Chan School of Public Health, Boston, MA (WF); Harvard University Center for African Studies, Boston, MA (WF); Heidelberg Institute of Global Health, Germany (TB), and the George Washington University Milken Institute of Public Health, Washington, DC (ES). The funders have no role in study design, data collection and analysis, decision to publish, or preparation of the manuscript."

Comment:

Response: We have included the statement in our cover letter as suggested. 

Comment:

Response: Due to issues with some national IRBs that are part of the ARISE network, some data cannot be availed on public domains. However, requests for data can be made by researchers directly to the national IRBs for access. We have provided the following additional information to clarify this issue.

“Individual participant data cannot be shared publicly. A data transfer agreement between Harvard T.H. Chan School of Public Health, Africa Academy for Public Health, and participating institutions (including Addis Continental Institute of Public Health, Nouna Health Research Center, Muhimbili University of Health and Allied Sciences, University of Dodoma, University of Ibadan, and Heidelberg Institute of Global Health) stipulates that data will be kept confidential and will not be shared beyond the research teams without prior permission. The de-identified dataset supporting this research may be made available following a request submitted to ghp@hsph.harvard.edu and be granted after obtaining permission from each participating institution.

Comment:

Response: Please see response above.

Comment:

Response: The ORCID ID for the corresponding author (IM) is included in the submission.

Comment:

Response: Our ethical statement is included in the Methods section as advised.

Reviewers' comments:

Reviewer's Responses to Questions

Comments to the Author

1. Is the manuscript technically sound, and do the data support the conclusions?

Reviewer #1: Yes

Reviewer #2: Yes

2. Has the statistical analysis been performed appropriately and rigorously? 

Reviewer #1: Yes

Reviewer #2: Yes

3. Have the authors made all data underlying the findings in their manuscript fully available?

Reviewer #1: No

Reviewer #2: Yes

4. Is the manuscript presented in an intelligible fashion and written in standard English?

Reviewer #1: Yes

Reviewer #2: Yes

5. Review Comments to the Author

Comment:

Reviewer #1: • I think the manuscript focused more on the diet quality rather than food price, as stated in Line 196. Therefore, I suggest to change the title to:

“The COVID-19 pandemic and its impacts on diet quality and food prices in sub-Saharan Africa”

 Response: We have changed the title of the manuscript as advised.

Comment:

• Line 69: in abstract, Dietary diversity score (DDS), should be mentioned in the Method used

Response: We have added the Dietary diversity score (DDS) to the methods section.

Comment:

• Line 69-70 could be done as one sentence, no need for repetition (data collected)

Response: We have removed the repeated text “ data collected”.

Comment:

• Line 76-84: there is no need to write the numbers here, as they are mentioned inside the text, additional study results could be stated here

Response: We thank the Reviewer for this suggestion. However, in our field of study of epidemiology, it is expected to include some numbers in the results section. We have edited the text to remove numbers when deemed not necessary.

Comment:

• Introduction: was written systematically

Response: We thank the Reviewer for this comment.

Comment:

• Line 122-125: Reference order would be 12-13-14, not 12-14-13

Response: We thank the Reviewer for noticing this. We have fixed this issue with references.

Comment:

• Line 150-151: All nine sites from the five countries (Figure 1) were included in the second round of data collection. (No need for repetition)

Response: We have deleted the statement as correctly noted by the Reviewer.

Comment:

• Line 160: Study design, is clearly described and detailed, still a sentence is needed to clearly stated the total sample size in each round as stated in line 273

Response: We thank the reviewer for this point. The samples collected in round 2 of the study for each site are shown in Table 1 page 14. Participation in round 1 and 2 of the survey is shown in the table below. We have included this table as a supplementary table 1 in the manuscript and added text on this.

“S Table 1 shows the number of participants in surveys 1 and 2 of the study.” (Line 386, page 9)

S Table 1: Proportion of participants participating in survey rounds

Proportion of participants participating in survey rounds Country/Region

 Burkina Faso

 Nouna Burkina

Faso

 Ouaga Ethiopia Addis Ethiopia Kersa Nigeria Ibadan Nigeria Lagos Tanzania 

Dar es Salaam Tanzania Dodoma Ghana Kintampo Total

Round 2 109

33.64

 125

41.53

 122

42.21

 117

39.26

 226

60.59

 129

44.48

 307

100.00

 347

100.00

 301

100.00

 1783

63.00 

Round 1 215

66.36

 176

58.47

 167

57.79

 181

60.74

 147

39.41

 161

55.52

 0

0.00

 0

0.00

 0

0.00

 1047

37.00

Total 324

11.45

 301

10.64

 289

10.21

 298

10.53

 373

13.18

 290

10.25

 307

10.85

 347

12.26

 301

10.64

 2830

100.00

Comment:

• Line 195 and 217: It is mentioned that the diet diversity score (DDS) was computed, still the DDS results are not shown on the results and discussion sections

Response: We thank the Reviewer for this observation. The findings from the DDS analysis were shown in the supplementary tables but not described in the methods section. We have included the findings of the analysis in the results section as shown below. As it is a secondary analysis we have not spend much time describing the findings in the discussion section. We hope this addresses the reviewer’s concerns.

“For the DDS food groups, the majority of respondents reported decreased consumption, with notable declines for dark green vegetables, other vegetables, staples, meats and fruits (S Fig 1). Further, the DDS for all sites at the different survey rounds are shown in S Fig 2. Overall DDS was lower in the first survey in 2020 and improved at the time of the second survey in 2021 in most sites, to levels closet to pre-COVID estimates. However, in Burkina Faso sites, in Kersa (Ethiopia) and Ibadan (Nigeria) DDS was lower in the second round compared to pre-COVID times (S Fig 2). (page 910-916, page 24)

Comments:

• Line 204-205: In addition to mention previous study using PDQS, it is recommended to mention some studies used the same dependent variables to compare results.

Response: We have included some additional studies that look at similar outcomes. However, because studies on diet quality have been limited in the African context, we refer to studies in other regions and we present this in the discussion section. Please see below additional text added.

“A study in Mexico also investigated the factors with diet quality early and later during the COVID-19 pandemic and found that consumption of healthy foods and food security declined, however consumption of unhealthy foods increased (35). In another study in the United States, individuals reporting food insecurity also lower consumption of both healthy and unhealthy foods (36). (lines 1009-1013, page 29)

Comment:

• Line 242: head of household (yes or no) what does (Yes) indicates? More explanation is needed

Response: The question referred to whether the respondent was the head of household. In the study context the household head may report better dietary intake than other household members, hence the need to consider this as a factor possibly influencing reported dietary intake.

• Line 246: Wealth index, more elaboration is needed of factor analysis

Response: We have provided additional clarification of the factor analysis approach.

“We computed the wealth index based on factor analysis on ownership of common household assets and other wealth-related indicators such as donkey cart, radio, television, bicycle, motorcycle ownership, access to grid electricity, improved water, fuel and roof within each country. We classified respondents in each country based into wealth tertiles based on results from the country specific factors selected.” (Lines 783-789, pages 12-13)

Comment:

• Line 252: Why GEE was adopted as an analytical tool? A brief justification is needed, with model specification

Response: We selected GEE linear regression as the study design included clustering. This use of this modeling approach is supported by literature (2).

Comments:

• Line 282 Table 1: for easy follow up and understanding, it is better to write only the percentage not numbers inside the Table

Response: We appreciate the concern of the reviewer on this. However, this is convention in epidemiology and we would expect that our audience would be interested in seeing both numbers. I hope this clarifies our decision to retain both the n (%).

Comments:

• Line 299-303 and line 239: These questions (worried about food, skipped a meal, gone for an entire day without eating.) are related to food consumption behaviors which is one of the types of food security measurement. Therefore, it is better to rephrase the sentences to reflect that. The same comment is applicable to Table 2

Response: We agree with the Reviewer. We have added the following sentence to clarify this for the audience.

“Food insecurity also affected respondents, with 45% reporting that they worried about food, and 30% said they had skipped a meal. Almost 10% had gone for an entire day without eating. These three questions are components of a metric for the assessment of food insecurity, the Household Food Insecurity Access Scale (HFIAS).”( Lines 838-841, page 17)

Comments:

• Line 314: Table 2:

- There is no need to repeat “Impact of COVID-19” on the subheading as it was reflected of Table title.

- Table title it is better to write five rather than 5.

Response: We have deleted the subheading and included “ five” in the table heading.

Comments:

- Is there any significance difference between these percentages???? may be another test is required

Response: We determined to use this table as a descriptive table. We decided that even if we show significant differences, without adjusting for possible confounders this may overemphasize findings erroneously. 

Comments:

• Line 324: All figures numbers should be revised to match the figures attached at the end of the manuscript

Response: We thank the Reviewer for noticing this error. We have fixed the figure headings to match last minute changes we had made.

Comments:

• Line 335: It is better to shorten the Fig 2 title

Response: We have changed the title for this figure 3 to “Fig 3. Changes in frequency of consumption of PDQS food groups comparing the pre-COVID-19 timepoint to the first and second surveys”

Comments:

• Lines 340 and 360: Also applicable to line 373

- “consumption of healthy PDQS food groups”, is it a right scientific terminology????

Response: We thank the reviewer for this question. We believe that this is correct terminology since the PDQS also assesses consumption of unhealthy foods/food groups as well.

Comments:

- Unit time is required with PDQS (week or 24 hrs)

Response: We often don’t include the frequency per week for the score. This information is captured in the consumption of the specific food groups.

Comments:

• Line 378-380: What is the source for data for” consumption of food groups at the time before the COVID-19 pandemic”, it should be stated in material and method section

Response: We have provided clarifying information on this in the manuscript.

“Respondents were asked to recall the number of days they consumed food from a list of 20 food groups over the past 7 days period before the COVID-19 emergency and during the COVID-19 pandemic (in the round 1 survey) (20) and in round 2 we asked respondents to recall their consumption of the same 20 food groups over the previous 7 days.” (Lines 713-717, page 10)

Comments:

• Line 608: “2020;4:1401-.” Please delete the dash (-) at the end

Response: Thank you for this. We have updated the citation.

Comments:

• Line 637: The publication year is missing

Response: We thank the reviewer for this. We have updated the citation number 37.

Comments:

Reviewer #2: The paper addresses an important recent topic and covered sub-researched countries in Africa. The writing is good and easy to follow; however, there is a need to improve some parts as discussed below. It would be also helpful if page number has been added to the document.

Introduction: The introduction should clearly state the main contributions of the study to strengthen the argument and ensure the importance of this work related to other relevant studies. A clearly stated and well-defined contribution is the merit of any research. Authors only indicated “Therefore, it is important that as the COVID-19 pandemic persists we continue to assess its impact on the nutrition and health of the households in SSA (12).” Then, the authors have details the strengths of the work on page 39 in the discussion. These strengths /contributions should also be indicated in the introduction.

Response: The Reviewer raises important points. We have addressed these comments by revising the introduction section as follows:

“We investigate the continued impacts of COVID-19 on diet quality and food prices, using data collected from the African Research, Implementation Science and Education (ARISE) Network cross-sectional COVID-19 studies by in five SSA countries, Burkina Faso, Ethiopia, Ghana, Nigeria and Tanzania. This study contributes to understanding the indirect impacts of COVID-19 on diets in a region where data is limited. Data from several countries allows us to understand the diverse pathways through which COVID-19 affected nutrition. We also have repeated cross-sectional studies in 3 countries, and this allows us to track changes in food security and diet intake at different times during the COVID-19 pandemic.” ( Lines 315-322, page 7) 

Comments:

Please add to the study design justification that the differences between the two samples/rounds will not cause issues in comparisons (across countries, across rounds if applicable) in the results section, especially that you conducted country-level differences in PDQS.

Response: We have added this information to the Methods section.

“We ensured consistency in study design and questions across all sites to ensure that differences between the two samples/rounds would not cause issues in comparisons (across countries and across rounds if applicable) in the analysis.” (Lines 653-655, page 9)

Comments:

Conclusion should include a summary of what the study is about, not only the outcomes.

Response: We have revised the conclusion section of the manuscript to incorporate this and other feedback from Reviewers.

Comments:

Minor comments are as follows:

Comments:

Table font is different than the text font and style.

Response: This is deliberate to ensure that the contents of the tables fit into the limited space within the manuscript.

Comments:

Texts in tables and figures should be in black to be more readable, it is in gray now.

Response: We have updated all text in tables to be in black font.

Comments:

Page 22: “Loss of or reduced salaries” I think you meant “lost or reduced salaries”

Response: We have updated the manuscript.

“Lost or reduced salaries were reported most in Ouagadougou (37%) and Lagos (26%).” ( Lines 828-829, page 16)

Comments:

Page 23 and 24: Not sure what do “(not shown) and “(results not shown)” mean? If you meant the results are not shown in the Table, please state that clearly.

Response: We mean that the findings are not shown in the tables or figures. We have revised all appropriate text to “results not shown in tables or figures”

6. PLOS authors have the option to publish the peer review history of their article (what does this mean?). If published, this will include your full peer review and any attached files.

Do you want your identity to be public for this peer review? For information about this choice, including consent withdrawal, please see our Privacy Policy.

Reviewer #1: No

Reviewer #2: No

 

References

1. ARISE Covid Survey Round 2 Methods Brief.

2. Pekár S, Brabec M. Generalized estimating equations: A pragmatic and flexible approach to the marginal GLM modelling of correlated data in the behavioural sciences. Ethology. 2018;124(2):86-93.

---

## [Decision Letter · Decision Letter 1]

12 Jun 2023

The COVID-19 pandemic and its  impacts on diet quality and food prices in sub-Saharan Africa

PONE-D-22-33938R1

Dear Dr. Madzorera,

We’re pleased to inform you that your manuscript has been judged scientifically suitable for publication and will be formally accepted for publication once it meets all outstanding technical requirements.

Kind regards,

Mohammed Al-Mahish

Academic Editor

PLOS ONE

Additional Editor Comments (optional):

Reviewers' comments:

Reviewer's Responses to Questions

**Comments to the Author**

1. If the authors have adequately addressed your comments raised in a previous round of review and you feel that this manuscript is now acceptable for publication, you may indicate that here to bypass the “Comments to the Author” section, enter your conflict of interest statement in the “Confidential to Editor” section, and submit your "Accept" recommendation.

Reviewer #1: All comments have been addressed

Reviewer #2: All comments have been addressed

2. Is the manuscript technically sound, and do the data support the conclusions?

Reviewer #1: Yes

Reviewer #2: Yes

3. Has the statistical analysis been performed appropriately and rigorously? 

Reviewer #1: Yes

Reviewer #2: Yes

4. Have the authors made all data underlying the findings in their manuscript fully available?

Reviewer #1: No

Reviewer #2: Yes

5. Is the manuscript presented in an intelligible fashion and written in standard English?

Reviewer #1: Yes

Reviewer #2: Yes

6. Review Comments to the Author

Reviewer #1: (No Response)

Reviewer #2: The authors have adequately addressed my comments raised in a previous round of review. Thank you for addressing the comments and suggestions.

7. PLOS authors have the option to publish the peer review history of their article (what does this mean?). If published, this will include your full peer review and any attached files.

Reviewer #1: No

Reviewer #2: No

---

## [Editor Report · Acceptance letter]

19 Jun 2023

PONE-D-22-33938R1 

The COVID-19 pandemic and its  impacts on diet quality and food prices in sub-Saharan Africa 

Dear Dr. Madzorera:

I'm pleased to inform you that your manuscript has been deemed suitable for publication in PLOS ONE. Congratulations! Your manuscript is now with our production department. 

Kind regards, 

on behalf of

Dr. Mohammed Al-Mahish 

Academic Editor

PLOS ONE